# LESS DATA LESS TOKENS: MULTILINGUAL UNIFICATION LEARNING FOR EFFICIENT TEST-TIME REASONING IN LLMs

## ABSTRACT

This paper explores the challenges of test-time scaling of large language models (LLMs), regarding both data and inference efficiency. We highlight the diversity of multilingual reasoning based on our pilot studies, and then introduce $L^2$, a multilingual unification learning approach with a decoding intervention strategy. The core idea of $L^2$ is that reasoning patterns vary across languages; leveraging this diversity can enhance both performance and efficiency. Specifically, we consider two forms of multilingual data: the entire long chain-of-thought annotations in different languages and stepwise mixtures of languages. By fine-tuning on them, we show that even small amounts of data can significantly improve reasoning capabilities. Our findings suggest that multilingual learning reduces both the required data and the number of inference tokens while maintaining comparable performance. Furthermore, $L^2$ is orthogonal to other data-efficient methods. Thus, we also emphasize the importance of diverse data selection. The $L^2$ method offers a promising solution to the challenges of data collection and test-time compute efficiency in LLMs.

## 1 INTRODUCTION

Scaling up training-time and test-time compute are two complementary strategies for enhancing the performance of large language models (LLMs). Training-time scaling allows the model to learn diverse knowledge from a massive corpus, but it often leads to unsatisfactory reasoning during inference, occasionally causing unreasonable errors. One explanation is that conventional inference primarily relies on pattern recognition from memory. In contrast, test-time scaling (e.g., OpenAI o1) significantly improves reasoning generalization by mirroring human cognitive processes, where problem solving is not always a direct input-to-output mapping as in supervised fine-tuning, but instead involves iterative reflection and error correction, with a longer thinking process (measured by the number of predicted tokens) guiding the model toward the correct answer.

A growing body of work has explored this idea, revealing two key challenges. The first is the heavy burden of data collection. Some attempts to replicate o1 require up to 747k training samples Guan et al. (2025), while DeepSeek-R1-32B necessitates 80k samples to achieve o1-level performance DeepSeek-AI et al. (2025). To reduce the costly long chain-of-thought (CoT) annotations, Sky-T1 distilled 17k samples from QwQ-32b Team (2025) using well-designed data selection strategies. S1 Muennighoff et al. (2025) further reduced the tuning dataset size to 1,000 by carefully selecting only high-quality, difficult, and diverse samples. Competition continues, with the latest work, LIMO Ye et al. (2025), demonstrating that as few as 817 samples can enable the model

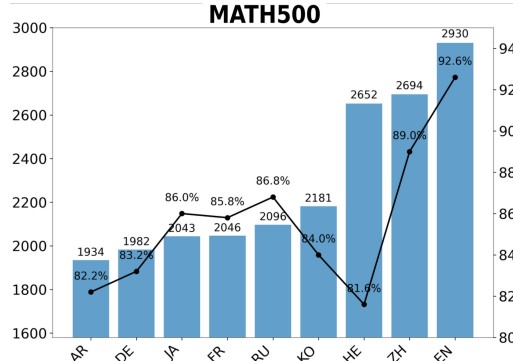

Figure 1: Pilot results of DeepSeek-R1-32B on MATH500 using different languages.

to acquire long reasoning capabilities and tackle highly challenging math problems. As the demand for annotations decreases, an interesting question arises: What is the limit of "less" data?

Another key challenge is the efficiency of test-time compute. As the reasoning chain expands, solving a problem often requires tens of thousands of tokens, significantly increasing the burden on inference efficiency. For ordinary problems, o1-type models use 1953% more tokens than traditional models to arrive at the same answer Chen et al. (2025). Higher performance on math competition problems often requires tens of thousands of tokens; thus, reducing inference tokens without sacrificing performance is crucial.

In this paper, we simplify the learning of test-time compute with **L**ess data and **L**ess inference tokens, namely $L^2$, through multilingual unification learning. Our core idea is that logical thinking varies across languages, leading to different solutions and inference token lengths given the same query. As shown in Figure 1, our pilot study translates English math questions into other languages, which are prompted to DeepSeek−R1−32B to seek solutions in their own languages. We observe that performance and efficiency vary substantially across languages. For example, on AIME24, accuracies range from 73.3% (French) to 40.0% (Hebrew), with inference tokens around 7k–9k (Section 2).

Therefore, we assume that augmenting a small amount of CoT data using multiple languages not only enhance data diversity, but also leverage the more concise thinking patterns in certain languages to improve inference efficiency.

To test our assumption, we propose a three-step $L^2$ multilingual unification learning framework: (1) collecting high-quality English samples (e.g., 6 from OpenAI o1, 1k from s1), (2) generating multilingual CoT annotations using the DeepSeek API, and (3) creating multilingual data by translating selected reflection steps and tagging them with language tokens; additionally, we introduce a decoding intervention strategy to guide language-specific inference.

We have conducted extensive experiments. Here are our main findings: **1)** Through data augmentation in different languages, only six high-quality samples can substantially improve long-reasoning performance. **2)** Multilingual enhancement is orthogonal to other learning strategies. By introducing more high-quality samples, the performance of our $L^2$-32B can be continuously improved, reaching comparable performance with 651 samples. **3)** While limited data can evoke extended reasoning, performance eventually plateaus; simply increasing samples or languages yields minimal gains, highlighting the need for more diverse data selection or construction. **4)** Multilingual learning enhances performance and notably reduces inference token usage compared to single-language learning. **5)** Once trained with multilingual data, it is unnecessary to infer with different languages. Our major contributions can be summarized as follows:

1. We highlight the differences in reasoning across languages, which not only helps enhance data diversity but also has the potential to improve reasoning efficiency.

2. We introduce $L^2$, a multilingual unification learning paradigm, which is orthogonal to other efficient data methods.

3. We constructed several datasets with different languages and scale. Based on them, we trained models to gain valuable insights for future research.

## 2 PRELIMINARY OBSERVATIONS ON MULTILINGUAL LONG REASONING

We begin by evaluating multilingual long CoT reasoning as in the pilot studies mentioned in the introduction. Specifically, we translate the AIME, GPQA, and MATH500 datasets into nine languages[2] and investigate how language choice affects accuracy, normal stopping rates, and token usage in each language. We also compare models of varying scales to examine the influence of multilingual factors on extended reasoning chains.

### 2.1 SETUP

To assess multilingual long-form CoT reasoning, we adopt a selection of open-source models varying in size and pretraining architecture, chosen for their demonstrated reasoning strength and suitability for local evaluation setups:

- **Qwen2.5-based Models** with parameter sizes of 1.5B, 7B, 14B, and 32B, including the Deepseek R1 Distilled Model, which is primarily trained on Chinese and English.

- **LLaMA-based Models** with parameter sizes of 8B and 70B, representing models pretrained on diverse multilingual corpora.

During inference, we record whether the model ends at an appropriate end-of-sequence marker (reporting the proportion of such "normal stops"), and we quantify the tokens generated in each language to assess whether reasoning genuinely unfolds in the target language. Due to space limitations, we report DeepSeek-R1-32B as a representative model given its strong performance; other results are provided in the Appendix. Note that scores are based on our careful reimplementation and may differ from prior reports due to varied prompts or other configurations.

## 2.2 OBSERVATION

Figure 2 summarizes the trends; we highlight three observations:

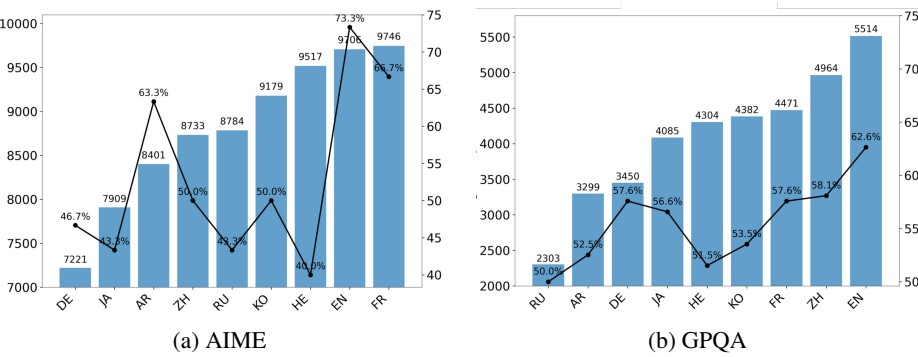

(a) AIME  (b) GPQA

Figure 2: Results of DeepSeek-R1-32B on AIME and GPQA using different languages. **Black polyline:** average completion tokens (right axis).

**Accuracy.** Our analysis indicates that English and Chinese achieve superior performance on GPQA and MATH500, consistent with their dominance in the pretraining corpora. Conversely, the AIME dataset shows notable exceptions: French, Hebrew, and Korean demonstrate unexpectedly competitive accuracies. We attribute these deviations primarily to AIME's limited size of only 30 problems, which may increase statistical variance and impact the stability of accuracy estimates.

**Normal Stopping and Token Usage.** Most outputs terminate correctly (though sometimes excessively or repetitively); however, token usage varies notably across languages.

**Multilingual Reasoning and Code-Switching.** For Chinese, English, and Korean prompts, the model predominantly reasons in that language; however, for other languages, the LLMs sometimes revert to Chinese or English midway, occasionally mixing languages within a single CoT.

In conclusion, the results indicate significant variations in accuracy and inference length across languages, suggesting distinct advantages. However, LLMs' occasional confusion between languages presents challenges for controlled multilingual reasoning, which will be discussed later.

## 3 DATA AND METHODS

To combine the merits of reasoning in different languages, our proposed $L^2$ multilingual unification learning first augments long CoT data at both the entire-solution level and the step level, then fine-tunes LLMs using the augmented data. The overall framework consists of three key steps: high-quality sample collection, multilingual thoughts annotation, and multilingual unification learning. Next, we introduce them in turn, followed by a multilingual decoding intervention to explore the impact of language on inference.

## 3.1 High-quality sample collection

**Notation.** For clarity, we denote training datasets as `[Source]-[Q]-[L]L`, where `Source` indicates the origin of the English questions, `Q` is the number of unique English questions, and `L` is the number of languages used for multilingual CoT augmentation. In this paper, `Source` $\in \{$`o1,S1,BS`$\}$, corresponding to OpenAI o1 examples, the S1k dataset, and Bespoke-Stratos-17k, respectively. For example, `o1-6-9L` denotes 6 English questions from o1 with CoT annotations in 9 languages. When referring to fine-tuned models, we prepend the base model. For example, $L^2$-`32B-S1-651-4L` denotes Qwen2.5-32B fine-tuned on the dataset `S1-651-4L`.

- **`o1-6-*L`.** This family contains six official examples adapted from OpenAI's website, manually curated and formatted in LaTeX. The topics include *Cipher, Coding, Math, Crossword, English, and Science*, with one question per topic. Here, the suffix "`-*L`" denotes the number of languages used for multilingual augmentation. For instance, `o1-6-4L` involves four languages (ZH, EN, KO, RU), resulting in 2,700 multilingual samples in total.
- **`S1-*L`.** We select 100, 651, and 1000 samples from the "S1k" dataset Muennighoff et al. (2025) to evaluate sample-size effects on training effectiveness. Initially, API instability restricted CoT generation, yielding `S1-651-4L`. Later, with API stability restored, the full 1k-sample set (`S1-1k-4L`) was completed. Experiments primarily use four languages (English, Chinese, Russian, Korean) to balance computational cost and comparative efficiency.

    **`BS-500-*`.** We randomly sample 500 questions from the Bespoke-Stratos-17k dataset Labs (2025), mainly covering mathematics and programming, to form an English seed set `BS-500`. Using the same multilingual CoT augmentation pipeline as for the `S1-*L` datasets, we add step-by-step CoT in four languages (EN, ZH, RU, KO), yielding `BS-500-4L`. In our experiments (Section 4.6), we further distinguish `BS-500-4L(full)` (full-solution CoT only) and `BS-500-4L(mix)` (full- plus step-level mixed-language CoT).

## 3.2 Multilingual thoughts annotation

We curate multilingual CoT at the solution level by translating questions with GPT-4o, generating step-by-step explanations via Deepseek API in target languages, and collecting diverse reasoning paths. *We manually verified translation quality for English and Chinese; other languages were not manually audited (see Limitations).*

## 3.3 Multilingual unification learning

We curate multilingual unification data by segmenting English CoT texts into reflection fragments, randomly translating selected steps (identified by cues like "Wait," "Hmm") via GPT-4o, and marking language boundaries with special tokens, thereby creating a code-switched corpus to foster flexible cross-lingual reasoning (illustrated in Figure 3).

**Training.** After the above two steps, we obtain the entire CoT in English and Chinese, respectively, as well as the stepwise mixture of thoughts in two languages.

We utilize the llamafactory framework, integrating flash attention and a light kernel acceleration package to expedite training. Our approach follows standard SFT with ZeRO Stage 3 optimization, and we set the maximum sequence length to 16k tokens. Training is conducted on 8 `H20` GPUs.

For datasets with fewer than 300 training samples (*small datasets*), we set batch size and gradient accumulation step to 1, oversample data to ensure sufficient coverage, and train until loss approaches zero. For larger datasets, we keep batch size at 1 but increase gradient accumulation step to 12 and train for 3 epochs.

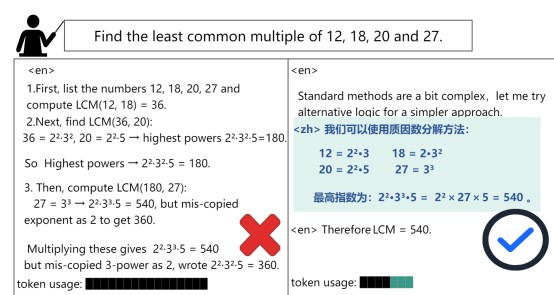

Figure 3: Mixed CN–EN reasoning uses fewer tokens with clearer logic.

### 3.4 DECODING INTERVENTION

We propose a *decoding intervention* during inference that adjusts language switching probabilities using special language tokens and hyperparameters. Specifically, given $\alpha \in [0, 1]$ controlling boost or suppression likelihood, magnitude $\beta$ for logit adjustment, and a top-$k$ cutoff, we sample $u \sim \text{Uniform}(0, 1)$ whenever a language token is within the top-$k$ candidates. If $u < \alpha$, we boost the token's logit by $+\beta$; otherwise, we penalize it by $-\beta$, thus shaping language usage.

---

**Algorithm 1: Decoding Intervention Pseudocode**

**Require:** logits, $\alpha, \beta, k$, zh_token
**Ensure:** adjusted_logits
1: $top\_tokens \leftarrow \text{TopK}(logits, k)$
2: **if** zh_token $\in top\_tokens$ **then**
3:     $u \leftarrow \text{Uniform}(0, 1)$
4:     **if** $u < \alpha$ **then**
5:         $logits[\text{zh\_token}] \leftarrow logits[\text{zh\_token}] + \beta$
6:     **else**
7:         $logits[\text{zh\_token}] \leftarrow logits[\text{zh\_token}] - \beta$
8:     **end if**
9: **end if**
10: **return** logits

---

## 4 EXPERIMENTS

This section details the experimental setup, baseline methods (Section 4.1), and key results, with a particular focus on performance under varying number of languages and data sizes.

### 4.1 BASELINES

To assess the effectiveness of our low-data multilingual long-chain-of-thought approach, we compare against several representative baselines:

- **OpenAI-o1** OpenAI (2024): A closed-source commercial large language model, the first to provide long-chain reasoning services.
- **Open-source models:** The base model Qwen2.5-32B Qwen et al. (2025), the QWQ model with the same 32B size, and the powerful O1-level open-source model, Deepseek R1 DeepSeek-AI et al. (2025).
- **Data-efficient models:** Sky-T1, s1, and LIMO, which were fine-tuned with as little as 17k, 1k, or even fewer examples Muennighoff et al. (2025); Ye et al. (2025), achieving performance comparable to o1-level models.

### 4.2 SETUP

We largely follow the experimental setup of s1 Muennighoff et al. (2025) for fair comparison. We choose `Qwen2.5-32B` as our base model and fine-tune it on three multilingual datasets introduced in Section 3.1: `o1-6-9L`, `S1-651-4L`, and `S1-1k-4L`. This yields three trained models: $L^2$-32B-`o1-6-9L`, $L^2$-32B-`S1-651-4L`, and $L^2$-32B-`S1-1k-4L`.

For assessment, we use the standard framework `vLLM` for inference with a temperature of 0.7, recording only the model's first response. Our evaluation covers three datasets: AIME24 (30), GPQA-Diamond (198), and MATH500 (500). We evaluate AIME and GPQA via string parsing and manually check decimals for MATH500.

### 4.3 MAIN RESULTS

Table 1 shows the overall results. We can see that:

**1)** With only 6 samples (augmented to 2,700), $L^2\text{--}32B\text{--}Mo1_6^{10}$ improves over the base model by **13.3%** (AIME24), **18.4%** (MATH500), and **8.5%** (GPQA), aligning with Table 1. **2)** With more high-quality data (i.e., 651 samples augmented to 4,500), we achieve performance comparable to models using much more data (e.g., **Sky-T1**, **Bespoke-32B**). This demonstrates the effectiveness of our multilingual unification learning. **3)** The strongest models are still those using much more data, like DeepSeek-R1 or o1.

Combined with the above conclusion, this suggests the importance of both curation of diverse data and how to select the high-quality ones.

Table 1: Overall performance on AIME24, MATH500 and GPQA-Diamond.

| Model | # ex. | AIME24 | MATH500 | GPQA |
|---|---|---|---|---|
| **API only** | | | | |
| o1-preview | N.A. | 44.6 | 85.5 | 73.3 |
| o1-mini | N.A. | 70.0 | 90.0 | 77.0 |
| o1 | N.A. | 74.4 | 94.8 | 77.3 |
| **Open weights** | | | | |
| Qwen2.5-32B | N.A. | 10.0 | 69.0 | 41.0 |
| QwQ-32B | N.A. | 50.0 | 90.6 | 65.2 |
| R1 | N.A. | 79.0 | 97.3 | 71.5 |
| R1-distill-7B | ∼800K | 72.0 | 94.3 | 62.1 |
| **Open weights + Open data** | | | | |
| Sky-T1 | 17K | 43.0 | 82.4 | 56.8 |
| Bespoke-32B | 17K | 63.0 | 93.0 | 58.1 |
| s1 w/o BF | 1K | 50.0 | 92.6 | 56.6 |
| s1-32B | 1K | 56.0 | 93.0 | 59.6 |
| LIMO | 1K | 57.1 | 94.8 | 66.7 |
| $L^2$-32B-o1-6-9L | 6 | 23.3 | 87.4 | 49.5 |
| $L^2$-32B-S1-651-4L | 651 | 63.3 | 93.0 | 60.0 |
| $L^2$-32B-S1-1k-4L | 1k | 63.3 | $95.0^3$/93.0 | 61.0 |

## 4.4 RQ1: How does extremely small training data affect test-time scaling?

In this experiment, we focus on the $\mathtt{o1-6-*L}$ family. Qwen2.5-32B is our base model. To ensure fair comparison, we first fine-tune it on the six English questions without CoT, obtaining a baseline model denoted $\mathtt{Qwen2.5-32B-o1-6-EN}$. Furthermore, $L^2$-32B-o1-6-1L denotes the model fine-tuned with only English CoT derived from DeepSeek R1, while $L^2$-32B-o1-6-4L and $L^2$-32B-o1-6-9L incorporate multilingual CoT in four and nine languages, respectively. As shown in Table 2, we conclude that:

Table 2: Accuracy on $L^2 - 32B - o1 - 6 - *L$ with multilingual augmentation.

| Setting | AIME | GPQA | MATH500 |
|---|---|---|---|
| $Qwen2.5\text{-}32B$ | 0.10 | 0.41 | 0.69 |
| $Qwen2.5\text{-}32B\text{-}o1-6-EN$ | 0.17 | 0.43 | 0.74 |
| $L^2\text{-}32B\text{-}o1-6-1L$ | 0.33 | 0.34 | 0.67 |
| $L^2$-32B-o1$-6-4L$ | **0.33** | **0.49** | 0.85 |
| $L^2$-32B-o1$-6-9L$ | 0.23 | **0.49** | **0.87** |

**1)** By tuning using six high-quality samples, even with some upsampling techniques, the baseline model $Qwen2.5\text{–}32B\text{–}o1\text{-}6\text{-}EN$ achieves only slight improvements. In contrast, our multilingual model $L^2\text{–}32B\text{–}o1\text{-}6\text{-}4L$ achieves significant gains across all datasets, demonstrating the effectiveness of the multilingual assumption in improving performance through increased data diversity.

**2)** Compared with $L^2\text{–}32B\text{–}o1\text{-}6\text{-}4L$, the improvement of $L^2\text{–}32B\text{–}o1\text{-}6\text{-}1L$ is much smaller. This indicates that even for the same questions, obtaining diverse reasoning data in multiple languages is crucial to enhance model performance.

## 4.5 RQ2: What is the upper limit of multilingual extension?

### 4.5.1 Analyzing the Impact of Data Scale

To investigate the impact of data scale on model performance, we randomly selected 100 questions from the S1 dataset as the initial query pool and constructed 10 incremental training datasets

---

[3] Based on manual inspection, some Math500 standard answers were incorrectly formatted, corresponded to multi-part fill-in answers, or involved decimals with inconsistent precision requirements. As a result, the format-based validator mistakenly flagged originally correct answers as wrong—affecting a non-negligible number of problems (8–12 out of 500; see the appendix for specific cases). The reported results have been corrected accordingly.

[2] Nine languages: Chinese (zh), English (en), French (fr), German (de), Arabic (ar), Hebrew (he), Japanese (ja), Korean (ko), Russian (ru).

`S1-Q-9L` with $Q \in \{10, 20, \ldots, 100\}$. For instance, the dataset labeled as `S1-10-9L` comprises 10 queries annotated with CoT reasoning in 9 different languages, as described in Section 3.1 using the Multilingual CoT method. Similarly, `S1-20-9L` is created by adding another 10 randomly selected queries from the same pool of 100 questions, ensuring no overlap with the previous 10 queries of `S1-10-9L`. This process is iteratively continued, expanding the dataset to include up to 100 queries and resulting in 10 datasets of increasing size $\{\texttt{S1-10-9L}, \ldots, \texttt{S1-100-9L}\}$. Each dataset is subsequently used to fine-tune a corresponding model $L^2$-32B-`S1-Q-9L`, and all models are evaluated under consistent experimental settings to ensure fair comparison.

The results demonstrate that around the scale of 30 queries, the model exhibits a distinct inflection point, where both its capabilities and token consumption increase significantly. This phenomenon was consistently observed across various evaluation datasets, including MATH500 (+45.8%), GPQA (+67.8%), AIME24 (+75.0%), and AIME25 (+175.0%) (Figure 4, Appendix). These findings suggest that a modest expansion of high-quality annotated data, particularly beyond the 30-query threshold, substantially enhances model performance by alleviating early-stage data scarcity and enabling the model to better generalize and leverage its reasoning capabilities.

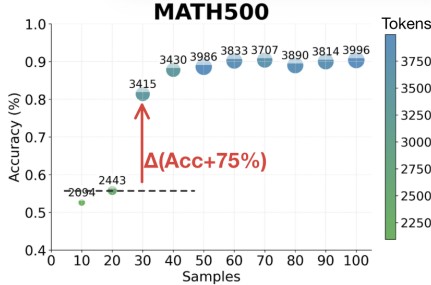

Figure 4: Accuracy vs training questions (point size/shade: tokens).

### 4.5.2 Evaluating Cross-Language Family Effects

We further investigated whether multilingual training across diverse language families improves model performance compared to training within a single language family.

We conducted the following experiment: the nine languages were grouped into three language families. As demonstrated in Section 4.5.1, training with 100 queries enables the model to develop long reasoning chains and improves performance across various datasets. For this experiment, we used the dataset `S1-100-9L`, which includes CoT reasoning in all nine languages. The corresponding model is denoted $L^2$-32B-`S1-100-9L`.

- **East-Asian:** Simplified Chinese (zh), Japanese (ja), Korean (ko)
- **Indo-European:** English (en), French (fr), German (de), Russian (ru)
- **Afro-Asiatic:** Arabic (ar), Hebrew (he)

In Figure 5, we generated training datasets by randomly combining different languages across these families and trained a model on each dataset. In the resulting visualization, each shape represents models trained with languages from specific language families. The more language families trained, the higher the accuracy and the fewer tokens used, yielding better results. Models positioned closer to the top-left corner indicate superior performance. Detailed numerical results can be found in the Appendix.

### 4.6 RQ3: Is our strategy orthogonal to other data curation methods?

Existing methods employ different strategies to select high-quality mathematical data. To further validate our approach, we increase the number of initial samples by randomly selecting samples from two typical sources: s1k and Bespoke-Stratos-17k Labs (2025). We have introduced the augmented s1k dataset `S1-651-4L` in Section 3.1, which consists of 651 math problems annotated with CoT in four languages (EN, ZH, RU, KO). For another source, we randomly select 500 samples from Bespoke-Stratos-17k, denoted `BS-500`, primarily featuring mathematics and programming problems.

After multilingual augmentation, we obtain the dataset `BS-500-4L(mix)`, which contains 500 multilingual CoT samples. When we remove the step of multilingual unification and instead train on `BS-500-4L(full)`, the model performance drops significantly. This suggests that the stepwise mixture of languages is crucial for enhancing generalization and reasoning capabilities.

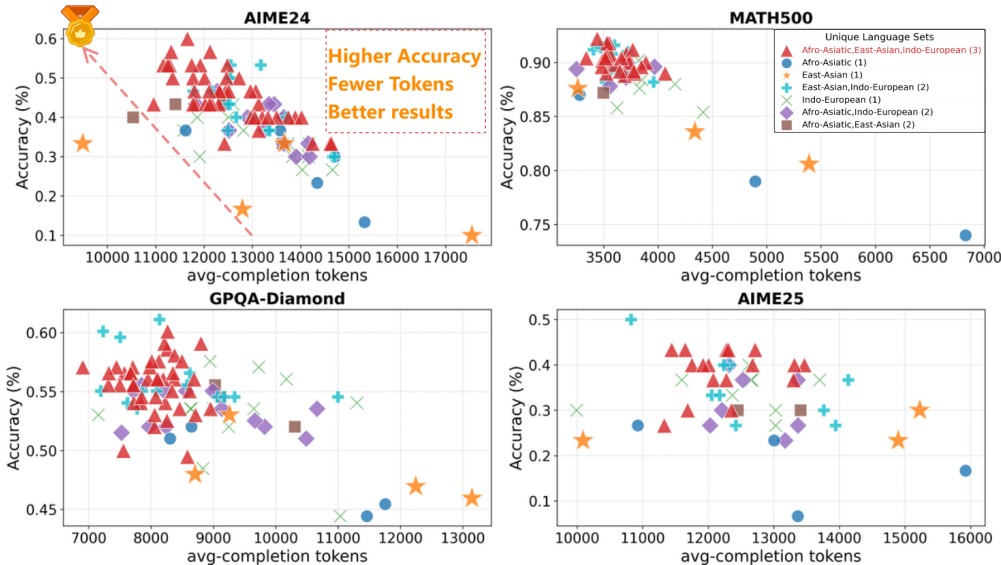

Figure 5: "East-Asian, Indo-European(2)" indicates a dataset including languages from both the East Asian and Indo-European families. Multiple shapes of the same kind indicate the same number of language families but with different combinations of specific languages.

We can see that regardless of the data source, our methods can effectively boost performance (Table 3). However, it is also noticeable that as the amount of original data increases, the marginal benefit of multilingual learning diminishes. This could be attributed to the model approaching its inherent capacity limits as the training data scale becomes larger.

## 4.7 RQ4: Does our strategy also improve inference efficiency?

We hypothesize that COT annotations from diverse language families offer complementary reasoning patterns, enhancing accuracy and inference efficiency through reduced token usage, unlike augmentations from linguistically similar sources, as shown in Figure 5 (see appendix).

## 4.8 RQ5:
### What happens if we control the reasoning language at decoding time?

In Figure 6, we guide models to express reflective reasoning in a specific language during decoding. In this setting, the maximum number of generated tokens increases from 15k in previous experiments to 131k. Using fine-tuned $L^2$ mod-

Table 3: Accuracy with BS-500 and S1-* datasets.

| Setting | AIME | GPQA | MATH500 |
|---|---|---|---|
| **BS-500 Data Set** | | | |
| $Qwen$2.5-32B-BS-500-EN | 0.43 | 0.52 | 0.90 |
| $L^2$-32B-BS-500-4L(full) | 0.46 | 0.55 | 0.91 |
| $L^2$-32B-BS-500-4L(mix) | **0.60** | 0.51 | **0.91** |
| **S1 Data Set** | | | |
| $Qwen$2.5-32B-S1-100-EN | 0.43 | 0.54 | 0.85 |
| $L^2$-32B-S1-100-4L | 0.53 | 0.53 | 0.90 |
| $Qwen$2.5-32B-S1-651-EN | 0.63 | 0.56 | 0.93 |
| $L^2$-32B-S1-651-4L | 0.63 | 0.60 | 0.93 |
| $Qwen$2.5-32B-S1-1k-EN | 0.60 | 0.60 | 0.91 |
| $L^2$-32B-S1-1k-4L | **0.63** | **0.61** | **0.93** |

els, we introduce an intervention ratio $\alpha$, controlling token frequency in the target language trained for reflection. This maintains reflective reasoning without sacrificing English fluency. We test varying $k \in \{2, 4, 6\}$; higher ratios encourage reasoning shifts (e.g., to Chinese). With $k = 2$ or $k = 4$, the model effectively switches languages, achieving **73.3%** accuracy on AIME24 (see Appendix). Token usage remains stable across most settings (App. B.3, Table 4), but at $k = 6$, excessive reflection interferes with standard reasoning.

## 5 Case Study

Figure 7 shows an example where the model solves a mathematical problem via multilingual reasoning. The model leverages multilingual capabilities to process and analyze the problem across languages; additional cases are provided in the Appendix. This approach allows the model to seamlessly process and analyze the mathematical problem across different languages, ensuring accurate and efficient solutions regardless of the language input. By leveraging the strengths of multi-language understanding, the model delivers robust and reliable responses in various linguistic contexts.

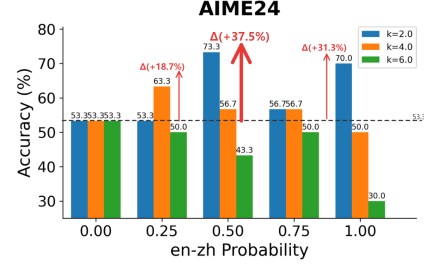

Figure 6: Decoding control on zh/en reasoning; results on AIME24.

## 6 RELATED WORK

### 6.1 TEST-TIME SCALING

Test-time scaling (TTS) enhances LLM performance by allocating extra computational resources during inference Liu et al. (2025); Zhang et al. (2025); Wu et al. (2025); Ji et al. (2025). Unlike traditional scaling methods, TTS enables fixed-parameter models to achieve superior outcomes through extended inference-time processing, akin to "thinking longer" Faria and Smith (2025). Diverse TTS strategies include computational budget control; sampling and search methods Muennighoff et al. (2025); Aggarwal and Welleck (2025); Son et al. (2025); verification-guided approaches Wang et al. (2025); Lifshitz et al. (2025); and latent-reasoning paradigms. Empirical results show significant reasoning gains, with smaller compute-optimal TTS models surpassing larger ones.

> A regular hexagon can be divided into six equilateral triangles. If the perimeter of one of the triangles is 21 inches, what is the perimeter, in inches, of the regular hexagon?

> **<en>**Okay, let's see. The problem says a regular hexagon can be divided into six equilateral triangles......Therefore, the perimeter of the hexagon is 42 inches. **<zh>**让我再核实一下。如果三角形的周长是 21 英寸，那么每条边就是 7 英寸。由于正六边形的每条边与这个三角形的边相等，因此六边形每条边的长度是 7 英寸。接着，六条边乘以 7 英寸得到 42 英寸。这看起来很直接。所以答案是 42 英寸。**<en>** so perimeter is 6*7=42. Yep, that's correct. Alternatively, maybe there's a different interpretation. **<zh>**因为每个三角形都是等边三角形，所以所有三条边的长度都应该相等。因此，这里的三条边应该都相等，也就是说，六边形的边长和半径的长度相等。所以，周长是三条边相加，即 3s = 21。因此，s = 7。由此可得，六边形的周长是 6s = 42。所以，答案仍然是 42。 Yes, that's the same result. So regardless of how I think about it, the answer is 42 inches. **Final Answer** The perimeter of the regular hexagon is \boxed{42} inches.

Figure 7: Case study: multilanguage reasoning for a math problem.

### 6.2 MULTILINGUALITY AND LOGICAL REASONING

Recent advances in large language models show that multilingual strategies significantly enhance logical reasoning Ghosh et al. (2025); Tran et al. (2025). While these models excel in high-resource languages like English, performance gaps persist for lower-resource languages Ravisankar et al. (2025). Techniques such as cross-lingual thought prompting (XLT) Huang et al. (2023) and English-pivoted CoT training exploit strong English reasoning to boost multilingual outcomes. Methods such as LayAlign Ruan et al. (2025) and AdaCoT Huang et al. (2025) further align abstract reasoning patterns across languages, promoting culturally responsive and globally applicable models.

## 7 CONCLUSION AND FUTURE WORK

In this paper, we present the $L^2$ approach, which leverages multilingual unification learning to enhance the test-time scaling of LLMs. Our method is demonstrated in incorporating a minimal amount of data and reducing the number of inference tokens, while maintaining long CoT reasoning capabilities. Our experimental results demonstrate that multilingual data can significantly improve long-reasoning tasks, with only a small number of high-quality samples yielding notable gains in performance. Furthermore, the $L^2$ approach offers a scalable and efficient path forward for training models that are capable of handling complex tasks while minimizing computational costs.

ETHICS STATEMENT

We use public benchmarks only: AIME24 (30 problems), GPQA-Diamond (198 items), and MATH500 (500 items). For extended experiments we draw from Bespoke-Stratos-17k and S1. Multilingual chain-of-thought (CoT) traces are produced by translating prompts with GPT-4o and generating step-wise rationales via the DeepSeek API, as detailed in §3.2. We attribute all datasets and models in our references and abide by their licenses and usage policies.

REPRODUCIBILITY STATEMENT

Unless stated otherwise, we fine-tune Qwen2.5-32B (open weights) with the LLaMA-Factory framework, Flash-Attention, and ZeRO Stage 3 SFT (§3.3). Maximum sequence length is 16k tokens.

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

## LLM ASSISTANCE DISCLOSURE

We used large language model (LLM) tools for grammar and wording refinement during manuscript preparation. All technical content, analyses, and citations were authored, verified, and remain the sole responsibility of the authors.

## A   LIMITATIONS

The $L^2$ approach offers promising efficiency for LLM test-time scaling but faces limitations, including varying language proficiency in base models and differences in tokenization due to linguistic variations, potentially affecting efficiency and results. Despite these, extensive experiments support our hypothesis. Integrating models trained on diverse languages also poses safety and quality risks, especially for low-resource languages, potentially causing biases and errors.

## B   ADDITIONAL RESULTS

### B.1   ACCURACY AND TOKEN CONSUMPTION ACROSS DIFFERENT MODELS AND LANGUAGES

Figures 8, 9, and 10 present detailed results illustrating the accuracy and token consumption of five language models—R1-Llama (8B, 70B) and R1-Qwen (1.5B, 7B, 14B)—evaluated across three benchmarks: AIME, GPQA, and MATH500.

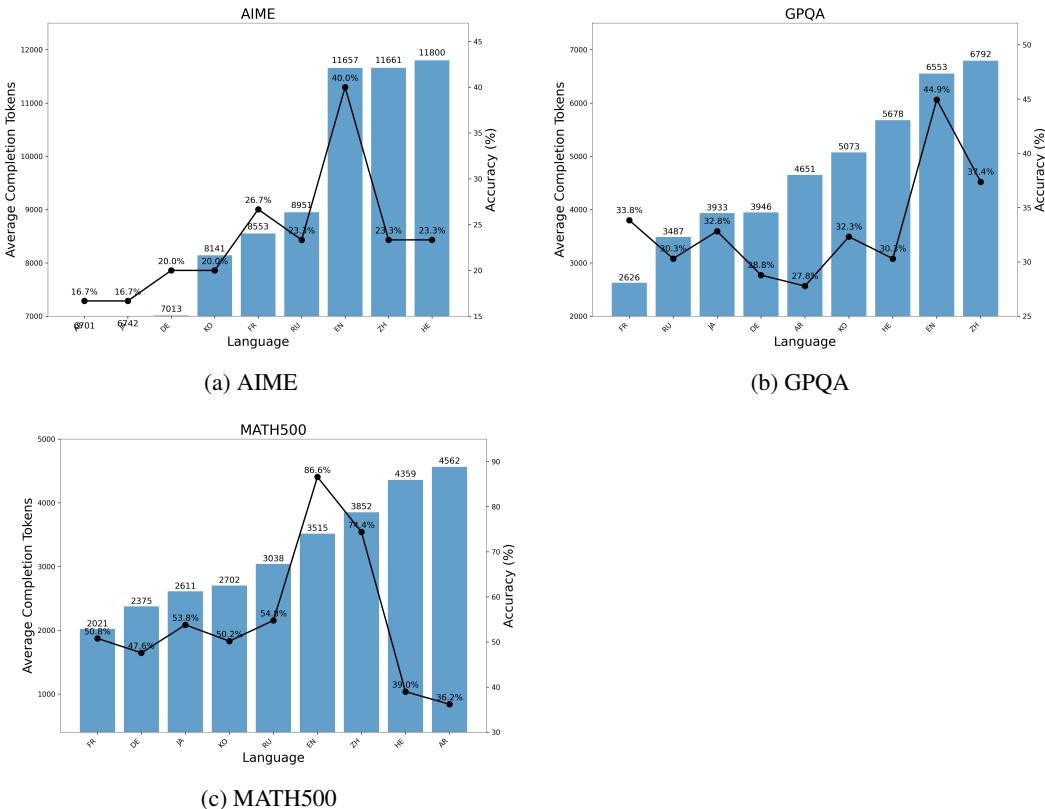

Figure 8: Results of R1-qwen-1.5b on AIME, GPQA, and MATH500 datasets using different languages.

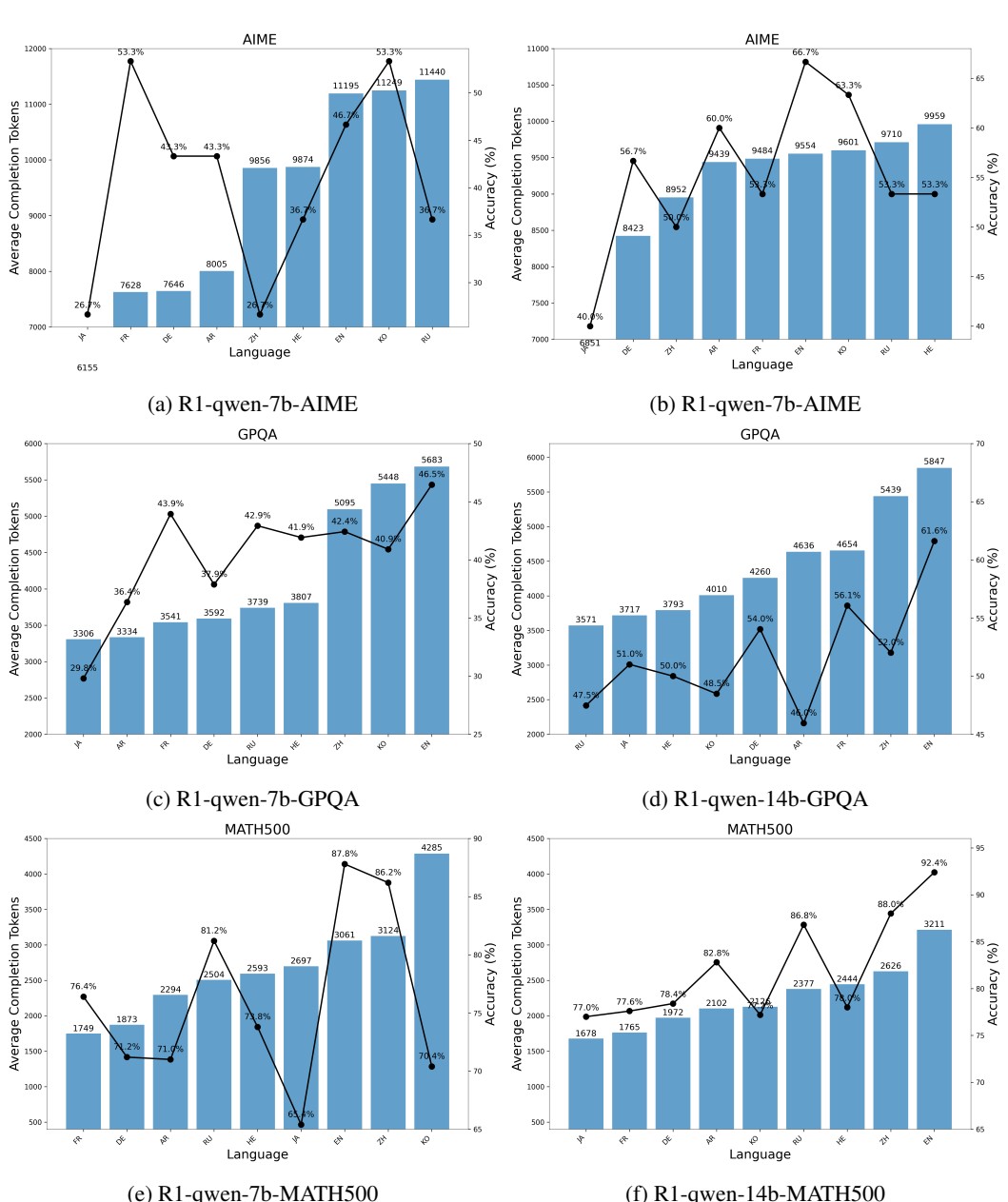

Figure 9: Results ofR1-qwen-7b andR1-qwen-14b on AIME, GPQA, and MATH500 datasets using different languages.

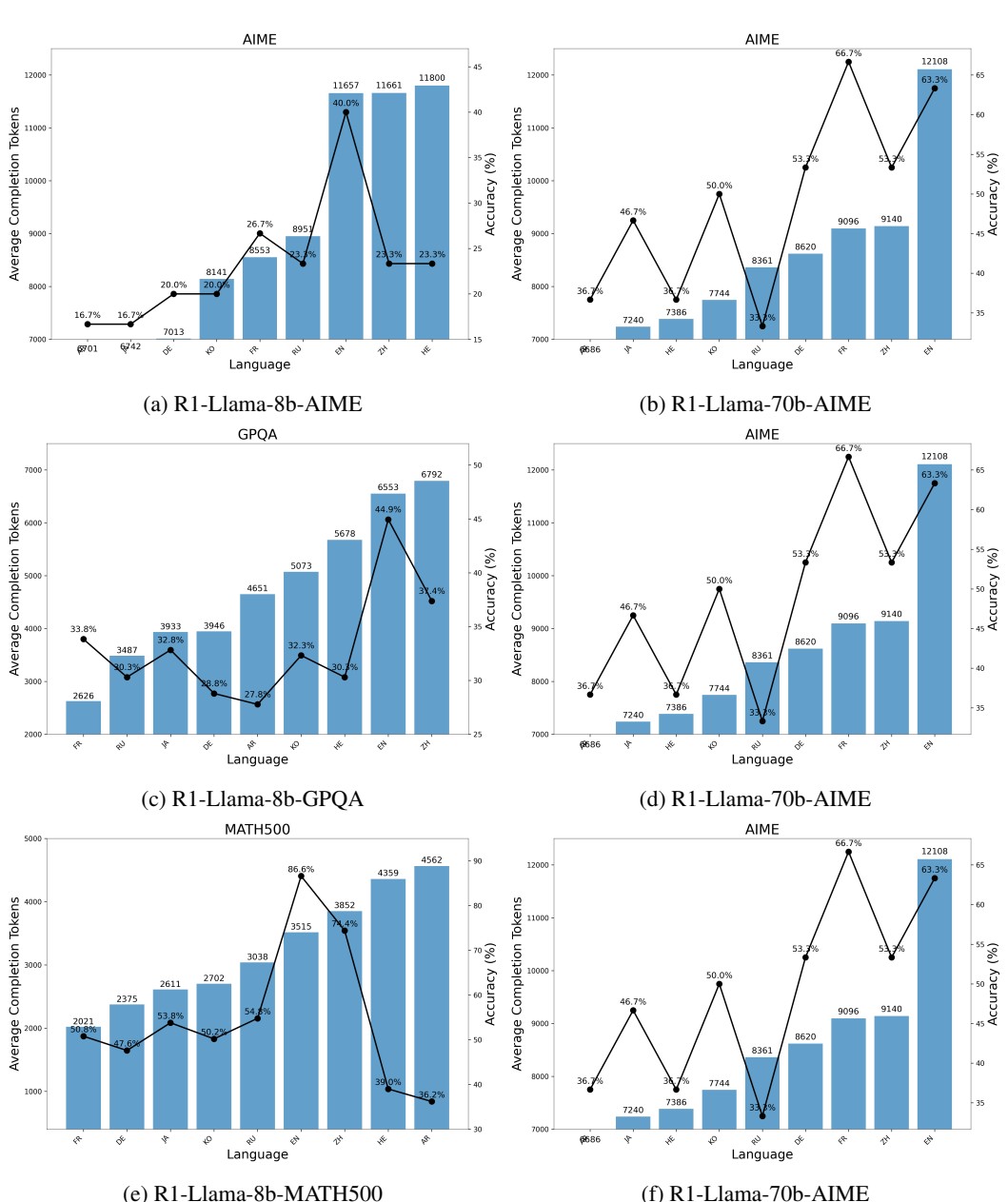

Figure 10: Results of R1-Llama-8b and R1-Llama-70b on AIME, GPQA, and MATH500 datasets using different languages.

## B.2 ACCURACY AND TOKEN CONSUMPTION ACROSS DIFFERENT MODELS AND LANGUAGES

Figure 11 comprehensively illustrates the relationship between the number of training samples, model accuracy, and generated tokens across the AIME24, AIME25, GPQA-D, and MATH500 benchmarks. Notably, there is a clear inflection point around 30 samples.

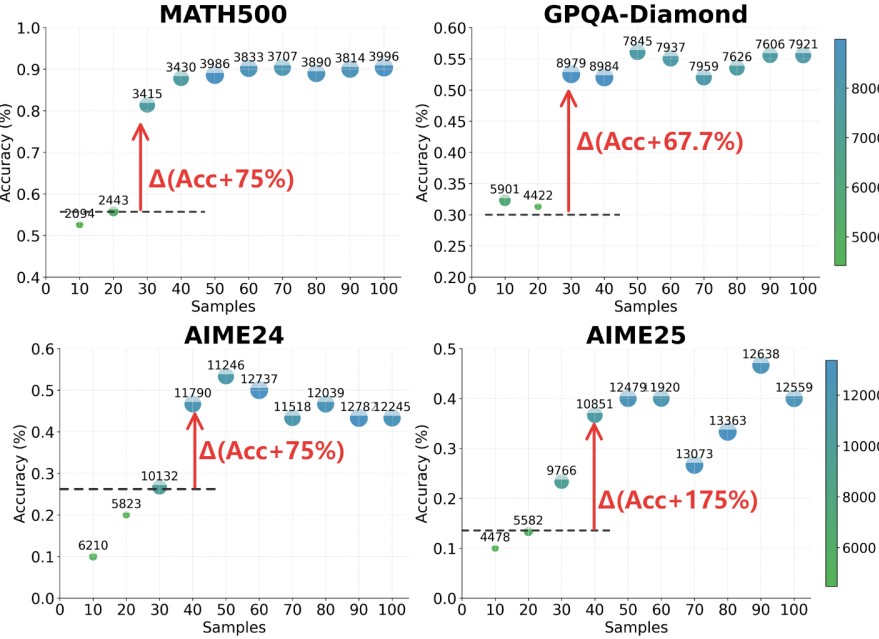

Figure 11: The x-axis indicates the number of questions included in the model training, and the y-axis denotes the achieved accuracy. Point size, shading intensity, and numeric annotations represent the quantity of generated tokens.

### B.3 SUPPLEMENTARY: TOKEN EFFICIENCY UNDER DECODING INTERVENTIONS

We quantify the trade-off between decoding interventions and efficiency. Table 4 reports accuracy and the average number of generated response tokens under different switching probabilities ($p_{\text{switch}}$) and the number of top-$K$ language candidates.

Table 4: Effect of switching probability ($p_{\text{switch}}$) and top-$K$ on accuracy and token cost. Tokens remain in a similar range and do not consistently increase as interventions intensify.

| $p_{\text{switch}}$ | $K$ | Accuracy (%) | Avg. Response Tokens |
|---|---|---|---|
| 0.00 | 2 | 53.33 | 13778.194 |
| 0.25 | 2 | 53.33 | 15516.871 |
| 0.50 | 2 | 73.33 | 14451.452 |
| 0.75 | 2 | 56.67 | 13770.355 |
| 1.00 | 2 | 70.00 | 13206.581 |
| 0.00 | 4 | 53.00 | 13778.194 |
| 0.25 | 4 | 63.33 | 12662.516 |
| 0.50 | 4 | 56.67 | 15642.355 |
| 0.74 | 4 | 56.67 | 12910.000 |
| 1.00 | 4 | 50.00 | 15500.419 |
| 0.00 | 6 | 53.33 | 13778.194 |
| 0.25 | 6 | 50.00 | 14704.000 |
| 0.50 | 6 | 43.33 | 12643.065 |
| 0.75 | 6 | 50.00 | 11956.903 |
| 1.00 | 6 | 30.00 | 14099.129 |

### B.4 CASE STUDY: MULTILINGUAL VS. MONOLINGUAL CORRECTION ON AIME24-I-6

To qualitatively illustrate how decoding interventions interact with reasoning, we compare two internal traces on the following AIME 2024 I Problem 6 variant:

> *Consider the paths of length* 16 *that follow the lines from the lower left corner to the upper right corner on an* $8 \times 8$ *grid. Find the number of such paths that change direction exactly four times.*

The correct answer to this problem is **294**.

In the multilingual reasoning with decoding interventions, the model initially reasons entirely in English and makes an incorrect structural assumption that the "number of direction changes equals the number of switches minus one." After several lines of English self-talk (*"Okay, let's see... Let me think..."*), it abruptly code-switches into Chinese and explicitly flags an error:

> **Multilingual trace (EN→ZH, excerpt)**
> *"Okay, let's see... I need to find the number of paths... the number of direction changes is the number of switches minus one..."* ...
> "等等，不对，让我再想一想。比如说，如果你有一个路径像这样：$RURU$，那么每次从$R$到$U$或从$U$到$R$的切换都算作一次方向改变。"

This Chinese interjection ("等等" / "wait a second") explicitly triggers a structural re-evaluation. The model re-checks concrete examples such as RURUR and URURU, correctly concludes that a path with exactly four direction changes must be decomposed into five alternating blocks ($RURUR$ or $URURU$), and then applies a standard stars-and-bars argument: splitting 8 R-steps into 3 positive blocks and 8 U-steps into 2 positive blocks, and symmetrically the other way around. This yields

$$21 \times 7 + 7 \times 21 = 294,$$

and the multilingual solution trace converges to the correct answer $\boxed{294}$.

In contrast, under the purely English decoding condition, the model exhibits a long stretch of monolingual self-corrections that never fully break away from the same flawed structural assumption:

**Monolingual trace (EN-only, excerpt)**
*"Okay, let's tackle this problem step by step..."* ...
*"Wait, maybe the number of direction changes is the number of switches minus one."* ...
*"Wait, let me re-examine this with a $2 \times 2$ grid..."* ...

Despite multiple "*Wait, let me rethink this*" moments, the monolingual trace repeatedly returns to counting only paths that start with $R$, effectively keeping the incorrect structural frame and ending with exactly half the true count, $\boxed{147}$, while overlooking the symmetric case starting with $U$.

This side-by-side case highlights that the Chinese discourse marker "等等" in the multilingual trace acts as a stronger intervention: it corresponds to a genuine structural reset and reformulation of the problem (from "switches minus one" to "five alternating blocks + stars and bars"), whereas the English "*Wait*" in the monolingual trace mostly produces local backtracking without escaping the original mistaken schema.

**Multilingual reasoning trace .**
<|begin_of_thought|> Okay, let's see. I need to find the number of paths of length 16 from the lower left corner to the upper right corner on an $8 \times 8$ grid, with the condition that the path changes direction exactly four times. Hmm, okay. Let me start by understanding the problem better. First, the grid is $8 \times 8$, so the starting point is $(0, 0)$ and the ending point is $(8, 8)$. A path of length 16 must consist of 8 right (R) moves and 8 up (U) moves, arranged in some order. The total number of such paths without any restrictions would be $C(16, 8) = 12870$, but we have the condition that the path must change direction exactly four times. So, I need to count only those paths which change direction four times.

But what exactly is a direction change here? In a path made up of R and U moves, a direction change occurs when you switch from moving right to moving up or vice versa. For example, a sequence like RRRU has one direction change from R to U; similarly, UUUURRR has one direction change from U to R. So the number of direction changes is the number of times the direction switches between R and U. Therefore, a path with exactly four direction changes has exactly four points where it switches from R to U or from U to R.

Let me think. If we imagine the path as a sequence of Rs and Us, then the number of direction changes is equal to the number of times the sequence alternates between R and U. For example, a path like RRRUUURRRUUUURRRUU would have direction changes wherever each switch occurs. Let me count: from R to U after the first three Rs, then U to R after the next three Us, then R to U after two Rs, then U to R after three Us, then R to U after two Rs。

等等，这可能太多了。嗯，也许我需要一个更通用的方法。等等，或许可以换一种方式来思考。在一个从$(0, 0)$ 到$(8, 8)$ 的路径中，每一步要么是向右（R），要么是向上（U）。要改变方向，我们就不能连续多次进行相同类型的移动。也就是说，如果你有一个向右的移动后跟着另一个向右的移动（即RR），这不会算作方向改变。只有当移动从R 转变为U 或从U 转变为R 时，才会算作一次方向改变。因此，整条路径中的总方向改变次数就是R 和U 之间的切换次数减一。...

Thus, the number of such paths is $\boxed{294}$. <|end_of_solution|>

**Monolingual reasoning trace.**
<|begin_of_thought|> Okay, so I have this geometry problem here. Let me try to parse it step by step. The question is about two rectangles, $ABCD$ and $EFGH$. They're positioned such that points $D, E, C, F$ are collinear. Also, points $A, D, H, G$ lie on a circle. The given lengths are $BC = 16$, $AB = 107$, $FG = 17$, and $EF = 184$. We need to find the length of $CE$.

First, let's visualize the problem. Since $ABCD$ and $EFGH$ are rectangles, all their angles are right angles, and opposite sides are equal. Let me note down the sides of each rectangle:

For rectangle $ABCD$: - $AB = 107$ (length) - $BC = 16$ (width)

So opposite sides $AD = BC = 16$, and $CD = AB = 107$.

For rectangle $EFGH$: - $EF = 184$ (length) - $FG = 17$ (width)

Thus, opposite sides $EH = FG = 17$, and $GH = EF = 184$.

The problem states that points $D, E, C, F$ are collinear. Hmm, so these four points lie on a straight line. Let me try to sketch this mentally. Rectangle $ABCD$ has points $A, B, C, D$. Rectangle $EFGH$ has points $E, F, G, H$. The collinearity of $D, E, C, F$ suggests that these points lie on a line in that order?

Wait, maybe not necessarily in that order. Let me check. Wait, $D$ is a vertex of rectangle $ABCD$. If $D$ and $C$ are adjacent vertices (since $ABCD$ is a rectangle), then $C$ is next to $D$. So maybe the collinearity is $D, E, C, F$? Or perhaps another order?

Let me think. If $D$ and $C$ are part of rectangle $ABCD$, then they are connected by side $DC$. But since $D, E, C, F$ are collinear, maybe the line passes through $D$, then $E$, then $C$, then $F$? But $E$ and $F$ are vertices of rectangle $EFGH$. Hmm, this might require a more precise diagram. Alternatively, maybe the collinearity is $D, C, E, F$? ...

Thus, the correct length of $CE$ is $\boxed{104}$. `<|end_of_solution|>`

## B.5 COMPARISON WITH CLASSICAL BACK-TRANSLATION

To further clarify whether our performance gains stem from multilingual diversity or merely generic data augmentation, we conducted a controlled experiment comparing our proposed multilingual-training method with a classical back-translation baseline.

We selected five languages, English (EN), Chinese (ZH), German (DE), Korean (KO), and Hebrew (HE) for augmentation. In the multilingual-training (ours) condition, each high-quality English long cot example was expanded directly into these five languages, and the resulting multilingual reasoning traces were used to train a single unified model.

In contrast, the thinking-en-backtrans baseline employed a standard back-translation pipeline. Specifically, each original English CoT example was translated into ZH, DE, KO, and HE, and subsequently back-translated into English. Only the final back-translated English versions were used in training, effectively increasing the volume of English training data but omitting direct multilingual reasoning exposure. All translations were obtained using the `Qwen-MT-PLUS` machine translation system accessed via its API.

Table 5 summarizes accuracy and average inference token usage across the AIME24, GPQA-D, and MATH500 benchmarks. Our multilingual-training method consistently achieves higher average accuracy and significantly reduces token usage compared to the back-translation baseline. These results clearly demonstrate that explicitly introducing multilingual reasoning diversity provides substantial benefits beyond simplistic data augmentation.

Table 5: Performance comparison between our multilingual-training method and a classical back-translation baseline. Reported metrics include accuracy and average inference token counts (lower is better).

| Method | AIME24 (Acc/Tokens) | GPQA-D (Acc/Tokens) | MATH500 (Acc/Tokens) | Average (Acc/Tokens) |
|---|---|---|---|---|
| multilingual-training (ours) | **0.4333**/**12998.355** | 0.5303/**8622.477** | **0.8960**/**4221.365** | **0.6198**/**8614.06** |
| thinking-en-backtrans | 0.3667/14055.419 | **0.5404**/11531.000 | 0.8660/5131.263 | 0.5910/10239.22 |

## B.6 ROBUSTNESS TO NOISY MULTILINGUAL CoT ANNOTATIONS

To evaluate the robustness of our method against varying levels of multilingual translation and CoT annotation quality, we created three distinct training sets differing only in annotation accuracy, while maintaining a consistent dataset size. Specifically, we employed an LLM-as-judge pipeline utilizing the `GLM-4.6` model, accessed via its public API, to verify whether the final answers of the generated multilingual data matched the original CoT answers from S1.

We construct three datasets, LabelAcc-30%, LabelAcc-60%, and LabelAcc-100%, which have identical numbers of training examples but different annotation accuracies, so that any performance differences can be attributed to annotation quality rather than data volume.

Table 6 presents the accuracies achieved on the AIME24, GPQA-Diamond, and MATH500 benchmarks.

## B.7 MULTILINGUAL POLYMATH RESULTS

To further analyze multilingual reasoning, we report detailed results on the PolyMath benchmark, which covers 18 languages and four difficulty levels (low, medium, high, and top) Wang et al. (2025).

Table 6: Performance robustness against varying quality levels of multilingual CoT annotations. *LabelAcc-x%* indicates the percentage of translations and CoTs validated as correct by `GLM-4.6`, with the total training data volume held constant across all conditions.

| Dataset | AIME24 | GPQA-D | MATH500 | Average |
|---|---|---|---|---|
| LabelAcc-30 | 0.4333 | **0.5909** | 0.908 | 0.644 |
| LabelAcc-60 | **0.5000** | 0.5546 | 0.902 | **0.652** |
| LabelAcc-100 | 0.4667 | 0.5404 | 0.904 | 0.637 |

We compare our multilingual-training model with Qwen2.5-32B-Instruct across all difficulty levels and languages.

Table 7 summarizes average accuracy at each difficulty level by aggregating over all 18 languages. Table 8 then provides a more fine-grained view, reporting PolyMath accuracy (%) for each individual language and difficulty level. In Table 8, each cell shows baseline and our model together with the relative gain in parentheses; positive gains are highlighted in red, while negative gains are shown in green. Finally, Figure 12 visualizes these multilingual results using four radar charts (one per difficulty level), providing an intuitive comparison between the baseline and our model across all languages.

Language abbreviations follow the PolyMath paper and use standard IETF language tags: EN (English), AR (Arabic), BN (Bengali), DE (German), ES (Spanish), FR (French), ID (Indonesian), IT (Italian), JA (Japanese), KO (Korean), MS (Malay), PT (Portuguese), RU (Russian), SW (Swahili), TE (Telugu), TH (Thai), VI (Vietnamese), and ZH (Chinese).

Table 7: Average PolyMath accuracy (%) across 18 languages at four difficulty levels.

| Model | polymath-low | polymath-medium | polymath-high | polymath-top |
|---|---|---|---|---|
| Qwen2.5-32B-Instruct | 85.51 | 24.13 | 9.91 | 6.89 |
| Ours | **88.89** | **55.47** | **36.04** | **15.87** |

Table 8: PolyMath accuracy (%) for each language and difficulty level. Each entry shows baseline (Qwen2.5-32B-Instruct) / ours, with relative gain in parentheses (red: positive, green: negative).

| Language | Low | Medium | High | Top |
|---|---|---|---|---|
| EN | 97.60/96.80 (−0.82) | 29.60/55.20 (+86.49) | 12.80/31.20 (+143.75) | 7.20/15.20 (+111.11) |
| AR | 91.20/90.40 (−0.88) | 21.60/56.00 (+159.26) | 12.00/36.80 (+206.67) | 7.20/21.60 (+200.00) |
| BN | 87.20/90.40 (+3.67) | 20.00/56.00 (+180.00) | 8.00/38.40 (+380.00) | 8.00/14.40 (+80.00) |
| DE | 86.40/87.20 (+0.93) | 26.40/56.00 (+112.12) | 9.60/36.80 (+283.33) | 8.80/17.60 (+100.00) |
| ES | 92.80/92.80 | 25.60/60.00 (+134.37) | 11.20/34.40 (+207.14) | 10.40/16.80 (+61.54) |
| FR | 83.20/88.80 (+6.73) | 28.00/57.60 (+105.71) | 12.80/36.80 (+187.50) | 6.40/18.40 (+187.50) |
| ID | 88.00/92.80 (+5.45) | 27.20/56.80 (+108.82) | 10.40/34.40 (+230.77) | 8.80/13.60 (+54.55) |
| IT | 90.40/96.00 (+6.19) | 27.20/56.00 (+105.88) | 11.20/38.40 (+242.86) | 14.40/15.20 (+5.56) |
| JA | 81.60/88.00 (+7.84) | 26.40/57.60 (+118.18) | 8.00/39.20 (+390.00) | 4.80/19.20 (+300.00) |
| KO | 87.20/89.60 (+2.75) | 25.60/54.40 (+112.50) | 10.40/37.60 (+261.54) | 6.40/14.40 (+125.00) |
| MS | 92.80/93.60 (+0.86) | 24.00/57.60 (+140.00) | 10.40/32.80 (+215.38) | 3.20/19.20 (+500.00) |
| PT | 88.80/94.40 (+6.31) | 32.00/56.80 (+77.50) | 12.00/40.80 (+240.00) | 6.40/12.80 (+100.00) |
| RU | 93.60/92.80 (−0.85) | 27.20/57.60 (+111.76) | 11.20/38.40 (+242.86) | 8.80/14.40 (+63.64) |
| SW | 52.00/68.80 (+32.31) | 15.20/47.20 (+210.53) | 5.60/28.00 (+400.00) | 2.40/9.60 (+300.00) |
| TE | 64.80/67.20 (+3.70) | 14.40/48.00 (+233.33) | 2.40/32.80 (+1266.67) | 1.60/13.60 (+750.00) |
| TH | 84.80/88.80 (+4.72) | 20.00/53.60 (+168.00) | 12.00/35.20 (+193.33) | 5.60/13.60 (+142.86) |
| VI | 88.80/92.00 (+3.60) | 20.00/56.00 (+180.00) | 9.60/36.00 (+275.00) | 7.20/15.20 (+111.11) |
| ZH | 88.00/89.60 (+1.82) | 24.00/56.00 (+133.33) | 8.80/40.80 (+363.64) | 6.40/20.80 (+225.00) |
| Overall | 85.51/88.89 (+3.95) | 24.13/55.47 (+129.88) | 9.91/36.04 (+263.67) | 6.89/15.87 (+130.33) |

Figure 12: Radar chart comparing Qwen2.5-32B-Instruct and our multilingual-training model across 18 languages at four difficulty levels (low, medium, high, and top).

