# Appendix: AIME CASE

Source: 2024_AIME_II_Problems/Problem_14

Let $b\ge 2$ be an integer. Call a positive integer $n$ $b\text-\textit{eautiful}$ if it has exactly two digits when expressed in base $b$ and these two digits sum to $\sqrt n$. For example, $81$ is $13\text-\textit{eautiful}$ because $81 = \underline{6} \ \underline{3}_{13}$ and $6 + 3 = \sqrt{81}$. Find the least integer $b\ge 2$ for which there are more than ten $b\text-\textit{eautiful}$ integers.

<|begin_of_thought|>

Okay, so I need to find the smallest base b (where b is at least 2) such that there are more than ten b-beautiful numbers. A number n is b-beautiful if it has exactly two digits in base b, and the sum of these two digits equals the square root of n. The example given is n = 81 being 13-beautiful because in base 13, it's written as 63, and 6 + 3 equals the square root of 81, which is 9.

嗯，我们先来分析这个问题。既然 n 在基底 b 下是一个两位数，那么它必须满足 $b$ 以内的最大值，也就是 $b^2 - 1$。等等，但在基底 $b$ 中，一个两位数可以表示为 $x \cdot b + y$，其中 $x$ 和 $y$ 是在 $0$ 到 $b - 1$ 范围内的数字，而 $x$ 不能为 $0$。因此，n 的取值范围是从 $b \cdot 1 + 0 = b$ 到 $(b - 1) \cdot b + (b - 1) = b^2 - 1$。不过实际上，由于 $x$ 和 $y$ 都不能超过 $b - 1$，因此 n 的最大值确实是 $b^2 - 1$。

But n has to be a perfect square because sqrt(n) must be an integer (since the digits sum to sqrt(n), and digits are integers). So n must be a perfect square, and in base b, it's a two-digit number. Therefore, the two-digit perfect squares in base b correspond to squares between b and b^2 - 1. Wait, but sqrt(n) would be between ceil(sqrt(b)) and floor(sqrt(b^2 - 1)) + 1? Wait, sqrt(b^2 - 1) is roughly b - 1/b, so the integer sqrt would be up to b - 1. So the squares we are considering are k^2 where k is from ceil(sqrt(b)) up to b - 1. Wait, but maybe it's easier to think that n is a two-digit number in base b, so n is between b and b^2 - 1, and n must be a perfect square. So k^2 is between b and b^2 - 1, so k is between ceil(sqrt(b)) and floor(sqrt(b^2 - 1)) which is b - 1. Therefore, the number of two-digit squares in base b is (b - 1) - ceil(sqrt(b)) + 1.

But not all of these squares are necessarily beautiful. To be beautiful, the sum of the two digits in base b must equal sqrt(n). So for each k from ceil(sqrt(b)) to b - 1, we need to check if k equals the sum of the digits of k^2 when written in base b.

So the problem reduces to finding the smallest b such that the number of k satisfying this condition is more than ten. So we need to find b where there are more than ten k's such that when you write k^2 in base b, the sum of its digits is k.

Let me formalize this. Let's denote the two digits of k^2 in base b as d1 and d2, so that k^2 = d1 * b + d2, where 0 < d1 < b and 0 <= d2 < b. Then, the condition is d1 + d2 = k.

Therefore, substituting d1 and d2 into the equation:

d1 + d2 = k

But since k^2 = d1 * b + d2, we can substitute d2 = k - d1 into this equation:

k^2 = d1 * b + (k - d1)

So:

k^2 = d1 * b + k - d1

Rearranging:

k^2 - k = d1 * b - d1

Factor out d1 on the right:

k^2 - k = d1*(b - 1)

Therefore:

d1 = (k^2 - k)/(b - 1)

So d1 must be an integer between 1 and b - 1 (since it's the first digit in base b). Therefore, (k^2 - k) must be divisible by (b - 1), and the result d1 must satisfy 1 <= d1 <= b - 1.

Moreover, since d2 = k - d1, and d2 must satisfy 0 <= d2 < b. So:

0 <= k - d1 < b

But since d1 = (k^2 - k)/(b - 1), substitute:

0 <= k - (k^2 - k)/(b - 1) < b

Multiply through by (b - 1) (which is positive, so inequalities stay the same):

0 <= (k*(b - 1)) - (k^2 - k) < b*(b - 1)

Simplify the left inequality:

k*(b - 1) - (k^2 - k) >= 0

k*(b - 1) - k^2 + k >= 0

Factor k:

k*(b - 1 + 1) - k^2 >= 0

Wait, k*(b - 1) + k - k^2 >= 0

That is:

k*b - k^2 >= 0

k*(b - k) >= 0

Since k is positive (as it's the square root of n, which is a two-digit number, so k >= ceil(sqrt(b))), and b - k must be non-negative. Wait, but since k can go up to b - 1, then b - k is at least 1. So actually, k*(b - k) >= 0 is always true because k and (b - k) are both positive. Therefore, the left inequality 0 <= ... is automatically satisfied.

Now, the right inequality:

(k*(b - 1) - (k^2 - k)) < b*(b - 1)

Simplify the left side:

k*(b - 1) - k^2 + k = k*b* - k - k^2 + k = k*b* - k^2

等等，$(k \cdot (b - 1) - (k^2 - k) = k \cdot b - k - k^2 + k = k \cdot b - k^2)$。

So the right inequality is:

k*b* - k^2 < b*(b - 1)

Rearranged:

- k^2 + b*k < b^2 - b

Multiply both sides by -1 (inequality flips):

k^2 - b*k > b^2 - b

But this seems complicated. 或许更好的方法是将其视为关于 $(k)$ 的二次不等式：

k^2 - b*k - (b^2 - b) < 0

嗯，也许可以重新整理一下：

k^2 - b*k + (b^2 - b) > 0

等等，我不太确定。让我再检查一下。

Starting from:

k^2 - b*k > - (b^2 - b)

Which is:

k^2 - b*k + (b^2 - b) > 0

But this seems like a quadratic in k. Let's consider:

k^2 - b*k + (b^2 - b) > 0

The discriminant of this quadratic is:

D = b^2 - 4*(b^2 - b) = b^2 - 4b^2 + 4b = -3b^2 + 4b

But since b >= 2, -3b^2 + 4b is negative for b >= 2 because -3b^2 + 4b < 0 when 3b^2 > 4b, which is true for b > 4/3. Since b >= 2 > 4/3, discriminant is negative. Therefore, the quadratic k^2 - b*k* + *(b^2 - b) is always positive for all k (since the coefficient of k^2 is positive and discriminant is negative). Therefore, the inequality k^2 - b*k* + *(b^2 -

b) > 0 is always true for all real k, which would imply that the original inequality k*b* - k^2 < *b*(b - 1) is always true.

Wait, but that can't be. Wait, perhaps my algebra is wrong.

等等，我们来一步步重新追踪一下。

原来的表达式是：

$$k \cdot b - k^2 < b \cdot (b - 1)$$

将其重新整理为：

$$-k^2 + b \cdot k - b^2 + b < 0$$

将所有项移到左侧，可以表示为：

$$-k^2 + b \cdot k - b^2 + b < 0$$

两边同时乘以 -1（注意不等号方向需要改变）：

$$k^2 - b \cdot k + b^2 - b > 0$$

这是一个关于 $k$ 的二次不等式。我们来计算判别式：

$$D = (-b)^2 - 4 \cdot 1 \cdot (b^2 - b) = b^2 - 4b^2 + 4b = -3b^2 + 4b$$

对于 $b \geq 2$，有：

$$-3b^2 + 4b \leq -12 + 8 = -4 < 0$$

因此，判别式为负数，这意味着二次函数 $k^2 - b \cdot k + b^2 - b$ 对于所有 $k$ 都是正的（因为二次项系数为正）。因此，无论 $k$ 取何值，该不等式 $k^2 - b \cdot k + b^2 - b > 0$ 恒成立。

由此可知，最初的不等式 $k \cdot b - k^2 < b \cdot (b - 1)$ 对于所有 $k$ 都成立。但这种情况似乎不太合理，因为如果 $k$ 足够大，比如 $k = b$，那么 $k \cdot b - k^2 = 0$，这显然小于 $b \cdot (b - 1)$。等等，但如果 $k$ 的取值范围是从 $\lceil \sqrt{b} \rceil$ 到 $b - 1$，那么 $k \leq b - 1$。因此，$k \cdot b - k^2$ 的最大值出现在 $k = \dfrac{b}{2}$ 时（因为这个表达式相当于 $-k^2 + b \cdot k$，这是一条开口向下的抛物线）。其最大值为 $\dfrac{b^2}{4}$。而 $b \cdot (b - 1)$ 等于 $b^2 - b$。当 $b \geq 2$ 时，显然有 $\dfrac{b^2}{4} < b^2 - b$，这可以通过以下方式验证：

\[

\frac{b^2}{4} < b^2 - b \implies 0 < \frac{3b^2}{4} - b \implies 0 < 3b^2 - 4b。

\]

对于 $b \geq 2$，有 $3b^2 - 4b \geq 12 - 8 = 4 > 0$。因此，不等式成立。由此可知，对于所有 $k$，包括 $k$ 在区间 $[\sqrt{b}, b-1]$ 内的所有值，该不等式 $k \cdot b - k^2 < b \cdot (b-1)$ 始终成立。因此，右侧的不等式总是成立的。

由此可以得出， $d_2 = k - d_1$ 总是满足 $0 \leq d_2 < b$。因此，我们只需要考虑 $d_1 = \dfrac{k^2 - k}{b-1}$ 是一个整数且满足 $1 \leq d_1 \leq b-1$ 的 $k$ 即可。

Therefore, the problem reduces to finding the number of integers k such that:

1. ceil(sqrt(b)) <= k <= b - 1

2. (k^2 - k) is divisible by (b - 1)

3. (k^2 - k)/(b - 1) is between 1 and b - 1 inclusive.

So the key is that (k^2 - k) must be divisible by (b - 1), and the result d1 must be a valid digit in base b.

Therefore, for each k in the range ceil(sqrt(b)) to b - 1, check if (k^2 - k) is divisible by (b - 1), and if the resulting d1 is between 1 and b - 1.

But since d1 = (k^2 - k)/(b - 1), and d1 must be at least 1, so (k^2 - k)/(b - 1) >= 1 => k^2 - k >= b - 1 => k^2 - k - b + 1 >= 0.

Similarly, d1 <= b - 1 => (k^2 - k)/(b - 1) <= b - 1 => k^2 - k <= (b - 1)^2.

Therefore, for each k, we have:

k^2 - k - b + 1 >= 0

and

k^2 - k <= (b - 1)^2

But perhaps these conditions are automatically satisfied given the range of k?

等等，我们来分析一下第一个条件：k^2 - k >= b - 1。因为 k 至少是 ceil(sqrt(b))，我们来看当 k = ceil(sqrt(b)) 时，k^2 - k 的值是多少。如果 k >= sqrt(b)，那么 k^2 >= b，因此 k^2 - k >= b - k。由于 k <= b - 1（因为 k 最大到 b - 1），所以 b - k >= 1。因此，k^2 - k >= b - k >= 1。但我们需要的是 k^2 - k >= b - 1。嗯，当 k = ceil(sqrt(b)) 时，k^2 - k >= b - 1 吗？

我们来看一个例子，令 b = 4。那么 ceil(sqrt(4)) = 2。此时 k = 2，则 k^2 - k = 4 - 2 = 2。而 b - 1 = 3。所以 2 >= 3 并不成立。因此，对于 b = 4，当 k = 2 时，k^2 - k = 2 < 3 = b - 1。因此，d1 = (2)/(4 - 1) = 2/3，这不是整数。所以 k = 2 不符合条件。因此，第一个条件 k^2 - k >= b - 1 并不总是满足，因此我们需要确保这一条件成立。

Similarly, the second condition k^2 - k <= (b - 1)^2. Let's check when k = b - 1. Then k^2 - k = (b - 1)^2 - (b - 1) = (b - 1)(b - 2). Which is equal to (b - 1)^2 - (b - 1). Wait, but (b - 1)^2 is (b^2 - 2b + 1). So k^2 - k = (b - 1)^2 - (b - 1) + (b - 1) = (b - 1)^2. Wait, no:

等一下，k^2 - k = (b - 1)^2 - (b - 1)？等等，(b - 1)^2 = b^2 - 2b + 1。而 (b - 1)^2 - (b - 1) = b^2 - 2b + 1 - b + 1 = b^2 - 3b + 2。但这并不是 k^2 - k。等等，或许我需要换一种思路。

Wait, when k = b - 1, k^2 - k = (b - 1)(b - 1 -1) = (b - 1)(b - 2) = b^2 - 3b + 2. On the other hand, (b - 1)^2 = b^2 - 2b + 1. So k^2 - k = b^2 - 3b + 2. Is this less than or equal to (b - 1)^2 = b^2 - 2b + 1? Let's compare:

b^2 - 3b + 2 <= b^2 - 2b + 1

Subtract b^2 from both sides:

- 3b + 2 <= -2b + 1

Add 2b to both sides:

- b + 2 <= 1

Subtract 2:

- b <= -1

Multiply by -1:

b >= 1

Which is always true since b >= 2. Therefore, k^2 - k <= (b - 1)^2 for all k <= b - 1 and b >= 2. Therefore, the second condition is always satisfied. Therefore, the only condition we need to check is that (k^2 - k) is divisible by (b - 1) and that d1 = (k^2 - k)/(b - 1) is at least 1. Wait, but earlier with b=4, k=2, d1=2/3 which is not an integer, so even if d1 is supposed to be at least 1, if it's not integer, it's invalid. Therefore, the main conditions are:

1. (k^2 - k) is divisible by (b - 1)
2. (k^2 - k)/(b - 1) >= 1

Which translates to k^2 - k >= b - 1.

Therefore, combining these, for each k from ceil(sqrt(b)) to b - 1, we need:

(k^2 - k) is divisible by (b - 1)

and

k^2 - k >= b - 1.

But since k >= ceil(sqrt(b)), let's check if k^2 - k >= b - 1.

If k >= sqrt(b), then k^2 >= b, so k^2 - k >= b - k. But since k >= 2 (because b >= 2, ceil(sqrt(b)) >= 2?), so if k >= 2, then b - k <= b - 2. Wait, but we need k^2 - k >= b - 1. For example, take b=4, ceil(sqrt(4))=2. Then k=2, k^2 -k=2, which is 2 >= 3? No. So in this case, k^2 -k=2 < b -1=3. So this condition is not satisfied. Therefore, even though k >= ceil(sqrt(b)), k^2 -k might still be less than b -1. Therefore, we need to ensure that k^2 -k >= b -1. Let's see when this happens.

Rearranged:

k^2 -k -b +1 >=0

This is a quadratic in k. Let's solve for k:

k^2 -k - (b -1) >=0

The roots of the equation k^2 -k - (b -1) =0 are:

k = [1 ± sqrt(1 + 4*(b -1))]/2 = [1 ± sqrt(4b -3)]/2

So the positive root is [1 + sqrt(4b -3)]/2. Therefore, k must be greater than or equal to this value for k^2 -k - (b -1) >=0.

Therefore, the minimal k required is ceil([1 + sqrt(4b -3)]/2 )

Therefore, the range of k is from max(ceil(sqrt(b)), ceil([1 + sqrt(4b -3)]/2 )) to b -1.

But this seems a bit complicated. Alternatively, perhaps it's easier to note that for each k, we can compute d1 = (k^2 -k)/(b -1). If d1 is an integer and 1 <= d1 <= b -1, and 0 <= d2 =k -d1 <b, then n=k^2 is b-beautiful.

But since d2 =k -d1, and d1=(k^2 -k)/(b -1), then d2 =k - (k^2 -k)/(b -1) = [k*(b -1) -k^2 +k]/(b -1) = [kb -k^2]/(b -1). *But we already saw that this is equal to (kb -k^2)/(b -1). Wait, but since n=k^2 is two digits in base b, then d1b + d2 =k^2. Therefore, d2 must be equal to k^2 -d1b. But d2 is also equal to k -d1. Therefore, from n=k^2 =d1b +d2, and d2=k -d1, so substituting gives k^2 =d1b + (k -d1) = d1*(b -1) +k. Therefore, d1=(k^2 -k)/(b -1).* So as before.

Therefore, the key is that (k^2 -k) must be divisible by (b -1), and d1=(k^2 -k)/(b -1) must be between 1 and b -1 inclusive, and d2=k -d1 must be between 0 and b -1 inclusive. But since d1=(k^2 -k)/(b -1), and d2=k -d1, then d2= (k*(b -1) -k^2 +k)/(b -1) = (kb -k^2)/(b -1). *But since n=k^2 is two digits, d2 must be less than b. However, since d2= (kb -k^2)/(b -1), we need to ensure that this is non-negative and less than b.* But as we saw earlier, the right inequality is always satisfied because the quadratic is always negative. Wait, perhaps I need to check that d2 is non-negative.

So d2 = (kb -k^2)/(b -1) >=0 => kb -k^2 >=0 => k*(b -k) >=0. Since k >0 and b -k >=0 (because k <=b -1), this is always true. Therefore, d2 is non-negative. And since d2 =k -d1, and d1 >=1, then d2 <=k -1 <b -1 (since k <=b -1). 等等，如果 k = b - 1，那么 d2 = (b - 1) - d1。而 d1 = (k² - k)/(b - 1) = [(b - 1)² - (b - 1)]/(b - 1) = [b² - 2b + 1 - b + 1]/(b - 1) = [b² - 3b + 2]/(b - 1) = (b - 1)(b - 2)/(b - 1) = b - 2。因此，d2 = (b - 1) - (b - 2) = 1，这个值小于 b。所以，d2 是有效的。

Therefore, the only conditions are:

1. (k^2 -k) is divisible by (b -1)

2. d1=(k^2 -k)/(b -1) is between 1 and b -1 inclusive.

Therefore, for each k from ceil(sqrt(b)) to b -1, check these two conditions. The number of such k's is the number of b-beautiful numbers.

Therefore, the problem reduces to finding the smallest b >=2 such that the number of k satisfying these conditions is more than 10.

Therefore, we need to find the minimal b where there are more than ten divisors (k^2 -k) of (b -1), with d1 in the valid range.

Alternatively, since (b -1) divides (k^2 -k), we can write (b -1) divides k(k -1). Since k and k -1 are consecutive integers, they are coprime, so (b -1) must divide either k or (k -1). Therefore, either k ≡0 mod (b -1) or k ≡1 mod (b -1). Therefore, for each k in the range ceil(sqrt(b)) to b -1, if k ≡0 or 1 mod (b -1), then (b -1) divides k(k -1), hence (b -1) divides (k^2 -k). Therefore, the possible k's are those congruent to 0 or 1 modulo (b -1). However, since k must be in the range ceil(sqrt(b)) to b -1, and d1=(k^2 -k)/(b -1) must be between 1 and b -1 inclusive.

Therefore, the number of such k's is equal to the number of integers k in [ceil(sqrt(b)), b -1] such that k ≡0 or 1 mod (b -1) and d1=(k^2 -k)/(b -1) <=b -1.

Wait, but (b -1) divides k(k -1). If k ≡0 mod (b -1), then k = m*(b -1). Similarly, if k ≡1 mod (b -1), then k =1 + m*(b -1). Therefore, since k is in [ceil(sqrt(b)), b -1], the possible values of m are such that m*(b -1) >= ceil(sqrt(b)) and 1 + m*(b -1) >= ceil(sqrt(b)), and both k's are less than b.

But since k must be less than b, for k = m*(b -1) <b, so m <b/(b -1). Since m is a positive integer, m=1. Similarly, for k=1 + m*(b -1) <b, then m*(b -1) <b -1, so m <(b -1)/(b -1)=1. Hence m=0. Therefore, only possible k's are:

If k ≡0 mod (b -1), then k=0 or b -1. But k cannot be 0 because k is at least ceil(sqrt(b)). So only possible k is b -1. But k=0 mod (b -1) would require k=0 or k=b -1. But k=0 is invalid, so k=b -1. For k =b -1, check if d1=( (b -1)^2 - (b -1) )/(b -1) = (b^2 -2b +1 -b +1)/(b -1) = (b^2 -3b +2)/(b -1) = (b -1)(b -2)/(b -1) =b -2. Which is between 1 and b -1. So d1 =b -2, d2 =k -d1 =b -1 - (b -2)=1. So n=(b -1)^2. Let's check if this is a two-digit number in base b. (b -1)^2 in base b is a two-digit number: the first digit is d1 =b -2, the second digit is 1. So yes, it's a two-digit number. So k=b -1 is always a solution.

Similarly, for k ≡1 mod (b -1), then k=1 + m*(b -1). Since k >= ceil(sqrt(b)), and k <b. So m=0 gives k=1, but 1 is only valid if ceil(sqrt(b)) <=1, which is only when b=2 (since sqrt(2)≈1.41, ceil is 2, but b >=2. 当 \( b=2 \) 时，\(\lceil \sqrt{2} \rceil = 2\)，因此 \( k=1 \) 不成立。当 \( b \geq 3 \) 时，\(\lceil \sqrt{b} \rceil \geq 3\)，所以 \( k=1 \) 也不成立。因此，只有 \( m=0 \) 时，\( k=1 \) 是可能的，但仅在 \( \lceil \sqrt{b} \rceil \leq 1 \) 的情况下才成立，这只有当 \( b=2 \) 时才有可能。然而，当 \( b=2 \) 时，\(\lceil \sqrt{2} \rceil=2\)，所以 \( k=1 \) 并不符合条件。因此，对于 \( k \equiv 1 \mod (b-1) \)，唯一可能的 \( k=1 \) 被排除在外，除非 \( b=2 \)。但即便如此，\( k=1 \) 也不符合条件。因此，\( k \equiv 1 \mod (b-1) \) 并没有在范围内的解。等等，或许我算错了。

等一下，如果 \( k \equiv 1 \mod (b - 1) \)，那么 \( k = 1 + m \cdot (b - 1) \)，其中 \( m \geq 0 \)。但 \( k \) 必须满足 \( \text{ceil}(\sqrt{b}) \) 到 \( b - 1 \) 之间。如果 \( m = 0 \)，那么 \( k = 1 \)。只有当 \( \text{ceil}(\sqrt{b}) \leq 1 \) 时，\( k = 1 \) 才有效，但这只有在 \( b = 2 \) 时才可能成立。然而，当 \( b = 2 \) 时，\( \text{ceil}(\sqrt{2}) = 2 \)，因此 \( k = 1 \) 并不符合条件。

如果 \( m = 1 \)，那么 \( k = 1 + (b - 1) = b \)。然而 \( k \) 必须小于 \( b \)，所以 \( k = b \) 超出了范围。

因此，对于 \( k \equiv 1 \mod (b - 1) \)，没有任何 \( k \) 位于范围内。所以只有 \( k = b - 1 \) 是符合条件的唯一 \( k \)。但这与之前的结论相矛盾。

等一下，我们来看一个例子：假设 $b = 5$。那么 $b - 1 = 4$。于是，$k$ 必须满足 $\text{ceil}(\sqrt{5}) \approx 3$，所以 $3 \leq k \leq 4$。接着检查 $k = 3$ 和 $k = 4$。

For k=3: (3^2 -3)/4 =6/4=1.5, not integer. So invalid.

For k=4: (16 -4)/4=12/4=3. So d1=3, d2=4 -3=1. So n=16=4^2. In base 5, 16 is 31, which sums to 4. So yes, that's valid. So here, only k=4 is valid. So in base 5, there's only one beautiful number. 等等，但根据前面的逻辑，k=b-1=4 是一个解。然而，k=4≡0 mod 4? 不，4≡0 mod 4。所以 4≡0 mod 4。因此，k=4 是一个解。但是，在这种情况下，k=4≡0 mod 4，所以它是一个解。然而，在 base 5 中，k=4 是唯一的解。因此，对于 base 5，只有一个美丽的数。

等等，这与之前的结论矛盾：之前我认为 $k \equiv 0 \mod (b - 1)$ 或 $k \equiv 1 \mod (b - 1)$。然而，当 $b = 5$ 时，$b - 1 = 4$，因此 $k \equiv 0 \mod 4$ 或 $k \equiv 1 \mod 4$。在范围 $[3, 4]$ 中，$k = 4 \equiv 0 \mod 4$，而 $k = 3 \equiv 3 \mod 4$。所以只有 $k = 4$ 满足条件。但当 $k = 4$ 时，$(k^2 - k) / 4 = (16 - 4)/4 = 12/4 = 3$，这是整数。因此，$k = 4$ 是一个有效的解。因此，在基数 $b = 5$ 中，只有一个美丽的数。

等等，但也许除了 $k = b - 1$ 之外，还有其他可能的 $k$ 值。我们来看 $b = 6$ 的情况。那么 $b - 1 = 5$。因此，$k$ 必须满足 $\lceil \sqrt{6} \rceil = 3$，所以 $3 \leq k \leq 5$。接下来检查 $k = 3, 4, 5$。

k=3: (9 -3)/5=6/5=1.2, not integer.

k=4: (16 -4)/5=12/5=2.4, not integer.

k=5: (25 -5)/5=20/5=4. So d1=4, d2=5 -4=1. So n=25. In base 6, 25 is 41, which sums to 5. So yes, sqrt(25)=5. So this is valid. So in base 6, only k=5 is valid. So again, only one beautiful number.

等等，这表明，只有 $k = b - 1$ 是有效的解。那么，为什么题目中会存在超过十个的"美丽"数呢？这好像不对。可能我的推理中存在错误。

等等，我们来看一个不同的例子。假设 $b = 8$。那么 $b - 1 = 7$。于是，$k$ 的范围是 $\lceil \sqrt{8} \rceil = 3$ 到 $7$。接下来检查 $k = 3, 4, 5, 6, 7$。

k=3: (9 -3)/7=6/7 ≈0.857, not integer.

k=4: (16 -4)/7=12/7≈1.714, not integer.

k=5: (25 -5)/7=20/7≈2.857, not integer.

k=6: (36 -6)/7=30/7≈4.285, not integer.

k=7: (49 -7)/7=42/7=6. So d1=6, d2=7 -6=1. So n=49=7^2. In base 8, 49 is 61, which sums to 7. Correct. So only k=7 is valid. So again, only one.

等等，这表明对于每个 $b$，只有 $k = b - 1$ 是有效的？如果是这样，那么每个 $b$ 只会有一个 $b$ -beautiful 数，但这与题目要求矛盾，题目中提到需要有超过十个 $b$ -beautiful 数。因此，我的推理一定存在错误。

等等，或许还有其他的 $k$ 满足条件 $(k^2 - k)$ 可以被 $(b - 1)$ 整除。之前我假设有 $k \equiv 0 \pmod{(b-1)}$ 或 $k \equiv 1 \pmod{(b-1)}$，但这种情况仅适用于 $(b-1)$ 是质数的时候。如果 $(b-1)$ 不是质数，那么可能会有更多的 $k$ 满足条件 $(k^2 - k)$ 能被 $(b-1)$ 整除。

Wait, for example, suppose b -1 is composite. Let's take b=7, so b -1=6. Then, since 6 divides k(k -1). Since k and k -1 are coprime, 6 must divide either k or k -1. Therefore, either k ≡0 mod 6 or k ≡1 mod 6. So in base 7, the possible k's are those congruent to 0 or 1 mod 6. The range for k is ceil(sqrt(7))=3 to 6. So k=3,4,5,6.

Check k=3: 3 mod6=3≠0 or1. So invalid.

k=4:4 mod6=4≠0 or1. Invalid.

k=5:5 mod6=5≠0 or1. Invalid.

k=6:6≡0 mod6. So valid. So k=6. Then d1=(36 -6)/6=30/6=5. So d1=5, d2=6 -5=1. So n=36=6^2. In base7, 36 is 5*7 +1=35+1=36. So 51_base7, which sums to 6. Correct. So only k=6 is valid here. So again, only one.

等等，即使 \( b-1 \) 是合数，满足条件的 \( k \) 似乎仍然是唯一的。这表明，对于每个基底 \( b \)，只有 \( k = b - 1 \) 是满足条件的。但这与题目描述矛盾，因为题目中提到 \( 81 \) 是 \( 13 \)-beautiful，且存在其他 \( b \)-beautiful 的数字。等等，也许我错在假设只有 \( k \equiv 0 \) 或 \( 1 \pmod{b-1} \) 才能满足条件。让我再仔细想一想。

If (b -1) divides k(k -1), since k and k -1 are coprime, (b -1) must divide either k or k -1. Therefore, either (b -1) divides k or (b -1) divides k -1. Therefore, k ≡0 mod (b -1) or k ≡1 mod (b -1). However, since k is in [ceil(sqrt(b)), b -1], the number of such k's is equal to the number of multiples of (b -1) in that interval (which is 1: k=b -1) plus the number of numbers congruent to1 mod (b -1) in that interval. Similarly, if (b -1) divides k -1, then k ≡1 mod (b -1). So numbers congruent to1 mod (b -1) in the interval.

But in the interval [ceil(sqrt(b)), b -1], how many numbers are congruent to0 or1 mod (b -1)?

For example, take b=10. Then b -1=9. The interval is ceil(sqrt(10))=4 to 9. So numbers 4,5,6,7,8,9.

Numbers congruent to0 mod9: 9. Numbers congruent to1 mod9: 1,10. But 1 is less than ceil(sqrt(10))=4, and 10 is equal to b=10, which is excluded. So only k=9≡0 mod9. So here, only k=9. Then d1=(81 -9)/9=72/9=8. So d1=8, d2=9 -8=1. So n=81=9^2. In base10, 81 is 81, which sums to 9. Correct. So only one beautiful number here.

等等，这表明在每个基数中只有一个美丽的数字。但题目中提到，需要找到一个 \( b \)，使得美丽的数字超过十个。这显然与我的推导相矛盾。因此，我的推理中一定存在某种错误。

Wait, let's take the example given: 81 is 13-beautiful. So in base13, 81 is written as 63. Let's check if 6 +3=9= sqrt(81). Correct. So k=9. So in base13, k=9. Let's check if (9^2 -9) is divisible by12. 81 -9=72. 72 divided by12 is6, which is integer. So d1=72/12=6, d2=9 -6=3. So yes, in base13, 81 is 6*13 +3=78 +3=81. So 63_base13. So digits 6 and3 sum to9. Therefore, this works. So here, k=9. For base13, b -1=12. So 9 is neither 0 nor1 mod12. So 9≡9 mod12. But (12) divides (9^2 -9)=72. Indeed, 12 divides72. So even though 9 is not congruent to0 or1 mod12, 12 divides72. Therefore, my earlier assumption that k must be ≡0 or1 mod(b -1) is wrong. Because (b -1) divides k(k -1) does not require k≡0 or1 mod(b -1). It only requires that (b -1) divides the product k(k -1). So (b -1) can divide k(k -1) even if it doesn't divide either k or k -1, as long as the product is divisible. For example, if (b -1) is composite, its factors could divide k and k -1 respectively.

Therefore, my previous reasoning that k must be ≡0 or1 mod(b -1) is incorrect. Therefore, the number of k's such that (b -1) divides k(k -1) is not limited to those congruent to0 or1 mod(b -1). Therefore, the number of such k's can

be higher. Therefore, perhaps for some bases b, there are multiple k's where (b -1) divides k(k -1). Therefore, the number of b-beautiful numbers can be more than one. Therefore, the problem is to find the minimal b where there are more than ten such k's.

Therefore, the approach should be:

For a given b, find all integers k in [ceil(sqrt(b)), b -1] such that (b -1) divides (k^2 -k). Then count these k's. We need to find the minimal b where this count exceeds ten.

But how to compute this efficiently? Since b can be large, maybe we need a formula or a pattern.

Alternatively, note that (b -1) divides (k^2 -k) can be written as k^2 ≡k mod(b -1). So k^2 ≡k mod(b -1). Therefore, k(k -1) ≡0 mod(b -1). So the number of k's in [ceil(sqrt(b)), b -1] satisfying this congruence.

This is equivalent to solving k(k -1) ≡0 mod(b -1) for k in the given range. The number of solutions to this congruence is equal to the number of integers k in [ceil(sqrt(b)), b -1] such that k ≡0 or1 mod(d) for some divisors d of (b -1). But this might be complicated.

Alternatively, note that for each k in [ceil(sqrt(b)), b -1], k(k -1) must be divisible by (b -1). This is equivalent to (b -1) divides k(k -1). So for each k, check if k(k -1) is divisible by (b -1). The number of such k's is the number of b-beautiful numbers.

Therefore, the problem reduces to, for each b, compute the number of k in [ceil(sqrt(b)), b -1] such that (b -1) divides k(k -1). We need the minimal b where this count exceeds ten.

To find such b, we can start from b=2 and increment b until the count exceeds ten. However, this might be time-consuming. Is there a pattern or a way to estimate how many k's satisfy this for a given b?

Note that (b -1) divides k(k -1) implies that for each prime power p^r dividing (b -1), p^r divides k(k -1). Since k and k -1 are coprime, for each prime p dividing (b -1), either p^r divides k or p^r divides (k -1). Therefore, the number of solutions k mod (b -1) is 2^ω, where ω is the number of distinct prime factors of (b -1). However, this is only if (b -1) is square-free. If (b -1) has square factors, the number of solutions can be less.

But in general, the number of solutions to k(k -1) ≡0 mod m is multiplicative over the prime powers in the factorization of m. So if m=∏p_i^{r_i}, then the number of solutions is ∏(d(p_i^{r_i})), where d(p^r)=2 if p=2, and 2 otherwise. Wait, no. For each prime power p^r dividing m, the number of solutions to k(k -1) ≡0 mod p^r is:

- If p=2, there are 1 or 3 solutions mod 8, but in general for higher powers, it's 2 solutions.

等等，对于奇素数 $p$，方程 $k(k - 1) \equiv 0 \pmod{p^r}$ 的解是 $k \equiv 0 \pmod{p^r}$ 或 $k \equiv 1 \pmod{p^r}$。因此，对于每个奇素数幂 $p^r$，方程有 $2$ 个解。

For p=2, if r >=3, then there are 4 solutions mod 8, but for r=1, 2 solutions, for r=2, 4 solutions. Wait, let's check:

For p=2:

If r=1: mod 2. The solutions are k≡0 or1 mod2. So 2 solutions.

If r=2: mod4. The solutions are k≡0 or1 mod4. So 2 solutions.

If r=3: mod8. The equations $k(k-1) \equiv 0 \mod 8$. The solutions are $k \equiv 0,1 \mod 8$. Because if k is even, k-1 is odd, so 8 divides k. If k is odd, k-1 is even, so 8 divides k-1. Wait, but 8 divides k or k-1. So solutions are $k \equiv 0$ or $1 \mod 8$. So 2 solutions. Wait, but 8 divides k(k-1). If k is even, then k must be divisible by 8, because k-1 is odd and can't contribute any factors of 2. Similarly, if k is odd, k-1 is even and must be divisible by 8. Therefore, solutions are $k \equiv 0$ or $1 \mod 8$. So 2 solutions. Similarly, for higher r, the number of solutions is 2. So for p=2, regardless of r, the number of solutions is 2.

Therefore, for a general modulus m, the number of solutions to $k(k-1) \equiv 0 \mod m$ is $2^\omega$, where $\omega$ is the number of distinct prime factors of m. However, this is only if m is square-free. If m has square factors, the number of solutions can be less. Wait, but in our case, m = b-1. So if (b-1) is square-free, the number of solutions mod (b-1) is $2^\omega$. Therefore, the number of solutions k in [0, b-1) is $2^\omega$. Therefore, the total number of solutions in [1, b-1] is $2^\omega$. However, we are interested in k in [ceil(sqrt(b)), b-1]. Therefore, the number of solutions is the number of residues mod (b-1) that are in [ceil(sqrt(b)), b-1] and satisfy $k \equiv 0$ or $1 \mod p^r$ for each prime power in (b-1).

But this is getting complicated. However, if (b-1) is a product of distinct primes (square-free), then the number of solutions mod (b-1) is $2^\omega$, where $\omega$ is the number of distinct prime factors. Therefore, the number of solutions in the entire range [0, b-1) is $2^\omega$. Therefore, if we can find b such that (b-1) has many distinct prime factors, then $2^\omega$ will be large, leading to more solutions. Therefore, perhaps choosing b such that (b-1) is a product of the first few primes will lead to a large number of solutions.

For example, let's take $(b-1)=2·3·5·7=210$. Then b=211. Then the number of solutions mod 210 is $2^4=16$. Therefore, in the interval [0,210), there are 16 solutions. Therefore, in [ceil(sqrt(211)),210], the number of solutions would be 16 minus those solutions less than ceil(sqrt(211))≈15. So 16 - number of solutions less than 15. If none of the solutions are less than 15, then we have 16 solutions. But since the solutions are congruent to 0 or 1 mod 2,3,5,7, which would spread out the residues. However, some of these solutions might be less than 15, but perhaps most are above. Therefore, such a b=211 might have many beautiful numbers. But we need to find the minimal b.

But since the problem asks for the least b where there are more than ten beautiful numbers, we need to find the smallest b such that the number of solutions to $k(k-1) \equiv 0 \mod(b-1)$ in [ceil(sqrt(b)), b-1] exceeds ten.

Therefore, to find such b, we can start from small b and check the count. However, given that the example given is 81 being 13-beautiful, and in base 13, k=9. Let's check for b=13. Then $(b-1)=12=2^2·3$. So the number of solutions mod 12 would be $2^2=4$? Wait, but 12 is not square-free. For modulus with repeated prime factors, the number of solutions can be different.

Wait, for modulus $m=12=2^2·3$. Let's compute the number of solutions to $k(k-1) \equiv 0 \mod 12$.

This requires that 4 divides k(k-1) and 3 divides k(k-1). Since k and k-1 are coprime:

For 4 divides k(k-1): Since k and k-1 are consecutive, one of them is even. But to have 4 divides k(k-1), either k is divisible by 4 or k-1 is divisible by 4. Because if k is even, then k-1 is odd, so 4 divides k. If k is odd, then k-1 is even, and we need 4 divides (k-1). Therefore, solutions mod 4 are $k \equiv 0$ or $1 \mod 4$.

Similarly, for 3 divides k(k-1): Either $k \equiv 0 \mod 3$ or $k \equiv 1 \mod 3$.

Therefore, combining these using the Chinese Remainder Theorem, the solutions mod12 are the combinations of the solutions mod4 and mod3.

For mod4: solutions k≡0,1 mod4.

For mod3: solutions k≡0,1 mod3.

Therefore, combining these, we have four combinations:

1. k≡0 mod4 and k≡0 mod3: solution is k≡0 mod12.

2. k≡0 mod4 and k≡1 mod3: solution is k≡8 mod12 (since 8≡0 mod4 and 8≡2 mod3? Wait, 8 mod3=2, so 8≡2≡-1 mod3. Wait, we need k≡0 mod4 and k≡1 mod3. Let's solve:

k ≡0 mod4

k ≡1 mod3

Let k=4m. Then 4m ≡1 mod3 => m ≡1 mod3. So m=3n +1. Therefore, k=4*(3n +1)=12n +4. Therefore, k≡4 mod12.

等等，4 mod 3 = 1，所以 4 ≡ 1 mod 3。由此可得，k = 4 满足条件。因此，解为 k ≡ 4 mod 12。

Similarly,

3. k≡1 mod4 and k≡0 mod3: k≡1 mod4, k≡0 mod3.

k=4m +1. Then 4m +1 ≡0 mod3 =>4m ≡-1 mod3 => m ≡2 mod3 (since 4≡1 mod3, so 1m ≡-1 mod3 =>m ≡2 mod3). Therefore, m=3n +2. Therefore, k=4(3n +2) +1=12n +9. Therefore, k≡9 mod12.

4. k≡1 mod4 and k≡1 mod3: k≡1 mod4, k≡1 mod3. Then k≡1 mod12 (since if k≡1 mod4 and mod3, then k≡1 mod lcm(4,3)=12).

Therefore, the four solutions mod12 are k≡0,1,4,9 mod12.

Therefore, in the interval [ceil(sqrt(13)),12], which is [4,12], the valid k's are:

k=4,5,6,7,8,9,10,11,12.

Check which of these are ≡0,1,4,9 mod12.

k=4:4≡4 mod12. Valid.

k=5:5≡5 mod12. Not valid.

等等，k=4 ≡ 4 mod12，符合条件。

k=5 ≡ 5 mod12，不符合条件。

k=6 ≡ 6 mod12，不符合条件。

k=7 ≡ 7 mod12，不符合条件。

k=8 ≡ 8 mod12，符合条件。

k=9 ≡ 9 mod12，符合条件。

k=10 ≡ 10 mod12，不符合条件。

k=11 ≡ 11 mod12，不符合条件。

k=12 ≡ 0 mod12，符合条件。

Therefore, the valid k's are4,8,9,12. Now check if these k's lead to valid d1=(k^2 -k)/12.

k=4:

(16 -4)/12=12/12=1. d1=1, d2=4 -1=3. So n=4^2=16. In base13,16 is1*13 +3=16. Digits1 and3 sum to4. Correct.

k=8:

(64 -8)/12=56/12≈4.666. Not integer. So invalid.

等等，56 并不能被 12 整除。等等，64 - 8 = 56。56 ÷ 12 ≈ 4.666…。这不是整数。所以 k=8 是无效的。但根据之前的计算，k=8 满足模 12 的条件。然而实际上，(k² - k) 必须能被 12 整除。因此，尽管 k=8 满足同余条件，但 56 并不能被 12 整除，因此 k=8 是无效的。这是一个问题。

等等，我们来检查一下当 \( k=8 \) 的情况：\( k \equiv 8 \mod 12 \)。那么 \( k(k-1) = 8 \cdot 7 = 56 \)。\( 56 \mod 12 = 56 - 4 \cdot 12 = 56 - 48 = 8 \)。因此，\( 56 \equiv 8 \mod 12 \)。所以，\( 56 \) 并不被 \( 12 \) 整除。然而，根据之前的分析，\( k \equiv 8 \mod 12 \) 应该是一个解。但这似乎与事实矛盾。

等等，也许我的方法存在错误。等一下，根据之前的分析，我们考虑的是模 4 和模 3 的条件。对于 \( k=8 \)：
\[
k \equiv 0 \mod 4 \text{（因为 \(8 \div 4 = 2\)，余数为 \(0\)）},
k - 1 = 7 \equiv 1 \mod 3 \text{（因为 \(7 \div 3 = 2\)，余数为 \(1\)）}。
\]
因此，\( k=8 \) 满足 \( k \equiv 0 \mod 4 \) 且 \( k \equiv 1 \mod 3 \) 的条件，这与我们之前计算的第 2 种情况相符。但是实际上，\( 8 \times 7 = 56 \)，而 \( 56 \) 显然不能被 \( 12 \) 整除。这是为什么？

等等，\( 4 \) 可以整除 \( 8 \)，而 \( 3 \) 可以整除 \( 7 \) 吗？\( 7 \) 不能被 \( 3 \) 整除，但 \( 7 \equiv 1 \mod 3 \)。等等，\( 3 \) 可以整除 \( k(k-1) \) 当且仅当 \( k \equiv 0 \mod 3 \) 或 \( k \equiv 1 \mod 3 \)。由于 \( k=8 \equiv 2 \mod 3 \)，\( k-1=7 \equiv 1 \mod 3 \)。因此，\( 3 \) 可以整除 \( k-1 \)。所以 \( 3 \) 确实可以整除 \( 7 \)，但这不可能，除非 \( 7 \) 能被 \( 3 \) 整除。显然，\( 7 \) 不能被 \( 3 \) 整除。等等，我的推理哪里出错了？

等等，\( k(k - 1) \equiv 0 \mod 3 \) 当且仅当 \( k \equiv 0 \mod 3 \) 或 \( k \equiv 1 \mod 3 \)。这是因为 3 是一个素数。所以，如果 \( k \equiv 2 \mod 3 \)，那么 \( k - 1 \equiv 1 \mod 3 \)，因此 \( k - 1 \equiv 1 \mod 3 \)，这意味着 \( 3 \) 可以整除 \( k - 1 \)。等等，不对。如果 \( k \equiv 2 \mod 3 \)，那么 \( k - 1 \equiv 1 \mod 3 \)，所以 \( 3 \) 并不能整除 \( k - 1 \)。等等，如果 \( k \equiv 1 \mod 3 \)，那么 \( k - 1 \equiv 0 \mod 3 \)，因此 \( 3 \) 可以整除 \( k - 1 \)。同样地，如果 \( k \equiv 0 \mod 3 \)，那么 \( 3 \) 可以整除 \( k \)。

因此，如果 \( k \equiv 2 \mod 3 \)，则 \( k(k - 1) \equiv 2 \cdot 1 = 2 \mod 3 \)，这不为 0。所以，\( k(k - 1) \equiv 0 \mod 3 \) 当且仅当 \( k \equiv 0 \) 或 \( 1 \mod 3 \)。于是，\( k=8 \equiv 2 \mod 3 \)。因此，\( 8 \cdot 7 \equiv 2 \cdot 1 = 2 \mod 3 \)，这说明 3 并不能整除 \( 8 \cdot 7 \)。但根据之前的条件，我们有 \( k \equiv 0 \mod 4 \) 且 \( k

$\equiv 1 \mod 3$)。然而，$k=8$ 满足 $0 \mod 4$，但 $8 \equiv 2 \mod 3$。因此，$k=8$ 并不满足 $1 \mod 3$ 的条件。等等，这与我们的 CRT 解 $k \equiv 8 \mod 12$ 矛盾了。

等等，我们来看一下。在情况 2 中，我们要求 $k \equiv 0 \mod 4$ 且 $k \equiv 1 \mod 3$。我们找到的解是 $k \equiv 4 \mod 12$。等等，让我们重新解这个问题。

我们设定 $k \equiv 0 \mod 4$ 且 $k \equiv 1 \mod 3$。那么 $k = 4m$，其中 $m$ 为某个整数。接着，$4m \equiv 1 \mod 3$。因为 $4 \equiv 1 \mod 3$，所以该式可以简化为 $1 \cdot m \equiv 1 \mod 3$。因此，$m \equiv 1 \mod 3$。由此得到 $m = 3n + 1$。代入后得到：

$$
k = 4 \cdot (3n + 1) = 12n + 4。
$$

因此，$k \equiv 4 \mod 12$。

所以，$k = 4, 16, 28, \dots$。因此，$k=4$ 是满足条件的，但 $k=8$ 并不是。等等，$k=8 \equiv 8 \mod 12$。所以 $8$ 并不符合 $4 \mod 12$ 的条件。

等等，我现在有些糊涂了。在之前的分析中，对于模 12 的解，我们得到了 $k \equiv 0,1,4,9 \mod 12$。那么 $k=4, 8, 9, 12$ 在区间 $[4,12]$ 中。然而，$k=8 \equiv 8 \mod 12$，并不在这些解的范围内。等等，实际上，$k=8 \equiv 8 \mod 12$，并不满足 $0,1,4,9 \mod 12$ 的条件。因此，之前的结论是正确的，解为 $k=4,9,12$。

等等，$k=12 \equiv 0 \mod 12$。但 $k=12$ 等于 $b-1=12$。接下来检查 $k=12$：

$$
(144 - 12)/12 = 132/12 = 11。
$$

因此，$d_1=11$，$d_2=12 - 11=1$。所以 $n=144=12^2$。在基底 13 中，$144$ 表示为 $11 \cdot 13 + 1=143 + 1=144$。数字 $11$ 和 $1$ 的和为 $12$，而 $\sqrt{144}=12$。因此，这确实是成立的。所以 $k=12$ 是有效的。

综上，对于 $b=13$，有效的 $k$ 值为 $4, 9, 12$。现在我们再检查 $k=9$：

$$
(81 - 9)/12 = 72/12 = 6。
$$

因此，$d_1=6$，$d_2=9 - 6=3$。所以 $n=81=9^2$。在基底 13 中，$81=6 \cdot 13 + 3=78 + 3=81$。数字 $6$ 和 $3$ 的和为 $9$，而 $\sqrt{81}=9$。因此，这也是正确的。

Similarly, k=4 as before.

Therefore, in base13, there are three beautiful numbers: 16,81,144, and also k=12. 等等，k=12 对应的 n=144。那么，k=12 表示 n=144。在基底13中，144写成111。等等，144除以13等于11余1，因此是11×13 + 1=143 +1=144。所以是11和1，即111。但 1+1=2，这并不等于12。等等，不对。再等等，d1=11，d2=1，所以 11 + 1=12。这确实等于 sqrt(144)=12。明白了。所以是正确的。

Therefore, in base13, the beautiful numbers are n=16,81,144. 等等，还有 k=9, 4, 和 12。因此，一共有四个美丽的数字。等等，k=4 对应的 n=16，k=9 对应的 n=81，k=12 对应的 n=144。所以总共有三个。等等，但在区间 [4,12] 中，k=4, 9, 12。因此，一共有三个数。所以 base13 中有三个美丽的数字。然而，题目中给出的例子是 81 是 13-美丽的，但这里我们找到了三个。等等，题目中说："例如，81 是 13-美丽的……"。因此，可能 base13 有三个这样的数字。

But the problem asks for the least b where there are more than ten such numbers. Therefore, we need to find a larger b. Therefore, perhaps we need to find a b where (b -1) has more prime factors, leading to more solutions.

Alternatively, note that if (b -1) is a product of the first n primes, then the number of solutions is 2^n. For example, if (b -1)=2·3·5·7·11=2310, then b=2311, and the number of solutions would be 2^5=32. Therefore, in the interval [ceil(sqrt(2311)),2310], the number of solutions would be 32 minus those less than ceil(sqrt(2311))≈48. So 32 - number of solutions less than48. If there are, say, 4 solutions less than48, then we have28 solutions. But this is speculative. However, this suggests that choosing b such that (b -1) is a product of many primes will lead to a higher number of solutions.

But to find the minimal b, we need to check values of b starting from b=2 upwards until we find more than ten solutions. However, this is time-consuming. Let's try to compute for some b.

Let's try b=30. Then (b -1)=29, which is prime. Therefore, the number of solutions mod29 is2. Therefore, in the interval [ceil(sqrt(30))=6,29], the solutions are k≡0 or1 mod29. But in this interval, k=29≡0 mod29, and k=1≡1 mod29. But k=1 is less than ceil(sqrt(30))=6, so only k=29. Therefore, only one solution. So n=29^2=841. In base30, 841 is 29·30 +1=870 +1=871, which is not correct. Wait, no. Wait, 29^2=841. In base30, 841 divided by30 is28·30=840, so 841=28*30 +1. Therefore, the digits are28 and1, which sum to29. sqrt(841)=29. Correct. So only one beautiful number here.

Similarly, let's try b=10. Then (b -1)=9=3^2. The number of solutions to k(k -1)≡0 mod9. Since 9=3^2, we need 9 divides k(k -1). Since k and k -1 are coprime, either 9 divides k or 9 divides k -1. Therefore, k≡0 or1 mod9. In the interval [ceil(sqrt(10))=4,9], the numbers are4,5,6,7,8,9. Among these, k=9≡0 mod9. So only k=9. Check k=9: (81 -9)/9=72/9=8. d1=8, d2=9 -8=1. So n=81=9^2. In base10, 81=8*10 +1. Digits8 and1 sum to9. Correct. So only one.

Next, try b=15. Then (b -1)=14=2*7. So solutions are k≡0,1 mod2 and k≡0,1 mod7. Combining, using Chinese Remainder Theorem:

Solutions mod14:

1. k≡0 mod2 and k≡0 mod7: k≡0 mod14

2. k≡0 mod2 and k≡1 mod7: k≡8 mod14 (since 8≡0 mod2, 8≡1 mod7)

3. k≡1 mod2 and k≡0 mod7: k≡7 mod14

等等，7 mod 2 = 1，对吗？因为 7 是奇数。所以 7 ≡ 1 mod 2。因此，k ≡ 7 mod 14。

4. k≡1 mod2 and k≡1 mod7: k≡1 mod14.

Therefore, solutions are k≡0,1,7,8 mod14. In the interval [ceil(sqrt(15))≈4,14], the numbers are4,5,6,7,8,9,10,11,12,13,14. Check which are≡0,1,7,8 mod14:

k=7≡7 mod14

k=8≡8 mod14

k=14≡0 mod14

k=1≡1 mod14 (but 1<4, so excluded)

k=7,8,14 are valid.

Check k=7:

(49 -7)/14=42/14=3. d1=3, d2=7 -3=4. n=49. In base15,49=3*15 +4=45 +4=49. Digits3 and4 sum to7. sqrt(49)=7. Correct.

k=8:

(64 -8)/14=56/14=4. d1=4, d2=8 -4=4. n=64. In base15,64=4*15 +4=60 +4=64. Digits4 and4 sum to8. sqrt(64)=8. Correct.

k=14:

(196 -14)/14=182/14=13. d1=13, d2=14 -13=1. n=196. In base15,196=13*15 +1=195 +1=196. Digits13 and1 sum to14. sqrt(196)=14. Correct.

Thus, in base15, there are three beautiful numbers:49,64,196. So three solutions. Still not enough.

Next, try b=22. Then (b -1)=21=3*7. Solutions mod21:

Solutions are combinations of solutions mod3 and mod7.

mod3: k≡0 or1 mod3

mod7: k≡0 or1 mod7

Thus, four combinations:

1.  k≡0 mod3 and k≡0 mod7: k≡0 mod21

等等，0 mod 3 是 0，0 mod 7 也是 0。因此，k ≡ 0 mod 21。

2.  k≡0 mod3 and k≡1 mod7: Solve k=3m=7n +1. k≡1 mod7 and k≡0 mod3. Let k=7n +1. Then 7n +1≡0 mod3 =>7n≡-1 mod3 => n≡-1 mod3 (since7≡1 mod3). So n≡2 mod3. Thus, n=3p +2. Then k=7*(3p +2)+1=21p +15. Therefore, k≡15 mod21.

3.  k≡1 mod3 and k≡0 mod7: k=3m +1=7n. So 3m +1=7n. Then 3m≡-1 mod7 =>3m≡6 mod7 =>m≡2 mod7. So m=7p +2. Then k=3*(7p +2)+1=21p +7. Thus, k≡7 mod21.

等等，3*(7p + 2) + 1 = 21p + 6 + 1 = 21p + 7。因此，k ≡ 7 mod 21。

4.  k≡1 mod3 and k≡1 mod7: k≡1 mod21.

Therefore, solutions are k≡0,1,7,15 mod21.

In the interval [ceil(sqrt(22))≈5,21], the numbers are5,...,21. Check k≡0,1,7,15,21 mod21.

k=7,15,21≡0,7,15 mod21.

Check k=7:

(49 -7)/21=42/21=2. d1=2, d2=7 -2=5. n=49. In base22,49=2*22 +5=44 +5=49. Digits2 and5 sum to7. Correct.

k=15:

(225 -15)/21=210/21=10. d1=10, d2=15 -10=5. n=225. In base22,225=10*22 +5=220 +5=225. Digits10 and5 sum to15. Correct.

k=21:

(441 -21)/21=420/21=20. d1=20, d2=21 -20=1. n=441. In base22,441=20*22 +1=440 +1=441. Digits20 and1 sum to21. Correct.

Thus, three beautiful numbers:49,225,441. Still only three.

Next, try b=10. Already did that. Let's try b=22. Hmm, same as above.

等等，也许我们可以尝试一个 $( b-1 )$ 是两个质数乘积的情况。我们来试一下 $( b=10 )$，但是 $( 9=3^2 )$。或者，$( b=14 )$：$( b-1=13 )$（质数）。那么在这种情况下，仅有两个解。

Alternatively, let's try b=21. Then (b -1)=20=2^2*5.

Solutions to k(k -1)≡0 mod20.

This requires that 4 divides k(k -1) and 5 divides k(k -1).

For 4 divides k(k -1): k≡0 or1 mod4.

For5 divides k(k -1):k≡0 or1 mod5.

Using Chinese Remainder Theorem:

Combine solutions mod4 and mod5.

Possible combinations:

  1.  k≡0 mod4 and k≡0 mod5: k≡0 mod20.

等等，0 mod 4 和 0 mod 5 的解是 k ≡ 0 mod 20。

  2.  k≡0 mod4 and k≡1 mod5.

Solve k=4m=5n +1. Then 4m≡1 mod5 =>4m≡1 mod5 =>m≡4^{-1} mod5. Since4*4=16≡1 mod5, so inverse of4 is4. Thus, m≡4*1=4 mod5. Thus, m=5p +4. Then k=4*(5p +4)=20p +16. Therefore, k≡16 mod20.

  3.  k≡1 mod4 and k≡0 mod5.

k=4m +1=5n. So 4m +1=5n. Then4m≡-1 mod5 =>4m≡4 mod5 =>m≡1 mod5. So m=5p +1. Then k=4*(5p +1)+1=20p +5. Therefore, k≡5 mod20.

4. k≡1 mod4 and k≡1 mod5.

k≡1 mod4 and k≡1 mod5 =>k≡1 mod20.

Thus, solutions are k≡0,1,5,16 mod20.

In the interval [ceil(sqrt(21))≈5,20], the numbers are5,...,20.

Check k≡0,1,5,16 mod20:

k=5≡5 mod20

k=16≡16 mod20

k=20≡0 mod20

k=1≡1 mod20 (but 1<5, excluded)

Thus, valid k=5,16,20.

Check k=5:

(25 -5)/20=20/20=1. d1=1, d2=5 -1=4. n=25. In base21,25=1*21 +4=25. Digits1 and4 sum to5. Correct.

k=16:

(256 -16)/20=240/20=12. d1=12, d2=16 -12=4. n=256. In base21,256=12*21 +4=252 +4=256. Digits12 and4 sum to16. Correct.

k=20:

(400 -20)/20=380/20=19. d1=19, d2=20 -19=1. n=400. In base21,400=19*21 +1=399 +1=400. Digits19 and1 sum to20. Correct.

Thus, three beautiful numbers:25,256,400. Still three.

嗯，也许可以尝试 $b = 32$。那么 $b - 1 = 31$，这是一个质数。因此，在模 $31$ 下，解的数量为 2。于是，在区间 $[ceil(\sqrt{32}) \approx 6, 31]$ 中，解为 $k = 31$。此时 $n = 31^2 = 961$。在以 32 为基数下，$961 = 30 \cdot 32 + 1 = 960 + 1 = 961$。其数字为 30 和 1，它们相加为 31。这是正确的。因此，仅有一个解。

Next, try b=16. Then (b -1)=15=3*5.

Solutions to k(k -1)≡0 mod15. So combinations of solutions mod3 and mod5.

mod3: k≡0 or1 mod3

mod5: k≡0 or1 mod5

Thus, four combinations:

1. k≡0 mod3 and k≡0 mod5: k≡0 mod15.
2. k≡0 mod3 and k≡1 mod5: Solve k=3m=5n +1. 3m≡1 mod5. 3*2=6≡1 mod5, so m≡2 mod5. m=5p +2. Then k=3(5p +2)=15p +6. Thus, k≡6 mod15.

等等，6 mod 3 = 0，6 mod 5 = 1。正确。

3. k≡1 mod3 and k≡0 mod5: Solve k=3m +1=5n. 3m +1≡0 mod5 =>3m≡-1≡4 mod5. Multiply both sides by2 (inverse of3 mod5 is2, since3*2=6≡1 mod5). So m≡8≡3 mod5. m=5p +3. Then k=3(5p +3)+1=15p +10. Thus, k≡10 mod15.

等等，10 mod 3 = 1，10 mod 5 = 0。正确。

4. k≡1 mod3 and k≡1 mod5: k≡1 mod15.

Thus, solutions are k≡0,1,6,10 mod15.

In the interval [ceil(sqrt(16))=4,15], numbers are4,...,15.

Check k≡0,1,6,10 mod15:

k=6,10,15≡6,10,0 mod15.

k=6:

(36 -6)/15=30/15=2. d1=2, d2=6 -2=4. n=36. In base16,36=2*16 +4=32 +4=36. Digits2 and4 sum to6. Correct.

等等，6² = 36。但是 6 等于 sqrt(36)。没错。

k=10:

(100 -10)/15=90/15=6. d1=6, d2=10 -6=4. n=100. In base16,100=6*16 +4=96 +4=100. Digits6 and4 sum to10. Correct.

等等，100=10²。所以 sqrt(100)=10。对的。

k=15:

(225 -15)/15=210/15=14. d1=14, d2=15 -14=1. n=225. In base16,225=14*16 +1=224 +1=225. Digits14 and1 sum to15. Correct.

Thus, three beautiful numbers:36,100,225. So three solutions.

嗯，看起来即使我们尝试更大的 $b$，每种情况下只找到了 3 个解。也许可以尝试 $b = 30$，但我们已经发现只有一个解了。等等，也许 $b = 7$：$b - 1 = 6 = 2 \cdot 3$。

Solutions to k(k -1)≡0 mod6. So combinations of solutions mod2 and mod3.

mod2: k≡0 or1 mod2

mod3: k≡0 or1 mod3

Thus, four combinations:

1. k≡0 mod2 and k≡0 mod3: k≡0 mod6.
2. k≡0 mod2 and k≡1 mod3: k≡4 mod6 (since 4≡0 mod2, 4≡1 mod3).
3. k≡1 mod2 and k≡0 mod3: k≡3 mod6.
4. k≡1 mod2 and k≡1 mod3: k≡1 mod6.

Thus, solutions are k≡0,1,3,4 mod6.

In the interval [ceil(sqrt(7))≈3,6], numbers are3,4,5,6.

Check k≡0,1,3,4 mod6:

k=3≡3 mod6

k=4≡4 mod6

k=6≡0 mod6

Check k=3:

(9 -3)/6=6/6=1. d1=1, d2=3 -1=2. n=9. In base7,9=1*7 +2=7 +2=9. Digits1 and2 sum to3. Correct.

k=4:

(16 -4)/6=12/6=2. d1=2, d2=4 -2=2. n=16. In base7,16=2*7 +2=14 +2=16. Digits2 and2 sum to4. Correct.

等等，sqrt(16)=4。对的。

k=6:

(36 -6)/6=30/6=5. d1=5, d2=6 -5=1. n=36. In base7,36=5*7 +1=35 +1=36. Digits5 and1 sum to6. Correct.

Thus, three beautiful numbers:9,16,36. So three solutions.

等等，这表明当

$$(b-1)$$

是两个质数的乘积时，解的数量是 4，但在区间

$$[\text{ceil}(\sqrt{b}), b-1]$$

中，可能会出现一些解被排除，因此最终只剩下 3 个解。

Therefore, perhaps to get more solutions, we need (b -1) to have more prime factors. Let's try a b where (b -1) is the product of the first four primes:2,3,5,7. Then (b -1)=2*3*5*7=210, so b=211.

Then the number of solutions mod210 is2^4=16. Therefore, in the interval [ceil(sqrt(211))≈15,210], the number of solutions would be16 minus those less than15. If none are less than15, then16 solutions. But let's check.

First, compute ceil(sqrt(211)): sqrt(211)≈14.52, so ceil is15. Therefore, the interval is15 to210. The solutions mod210 are k≡0,1 mod2,3,5,7. Wait, no. Wait, the solutions are combinations of k≡0 or1 mod each prime factor. Since210=2*3*5*7, the number of solutions is2^4=16. Each solution corresponds to choosing for each prime p, k≡0 or1 modp. Then combine them using Chinese Remainder Theorem.

Each combination gives a unique solution mod210. Therefore, there are16 solutions mod210. Therefore, in the interval [0,210), there are16 solutions. Therefore, in the interval [15,210], the number of solutions is16 minus the

number of solutions less than15.

Check which of the16 solutions are less than15.

The16 solutions mod210 are:

For each prime factor2,3,5,7, choose k≡0 or1 modp. Then combine.

For example:

1. k≡0 mod2,3,5,7: k≡0 mod210. So k=0 mod210. In [0,210), this is0. But0 is excluded since k>=ceil(sqrt(211))=15. So excluded.

2. k≡0 mod2,3,5 and≡1 mod7: Solve k≡0 mod(2*3*5)=30 and k≡1 mod7. So k=30m≡1 mod7. 30≡2 mod7, so 2m≡1 mod7. Multiply both sides by inverse of2 mod7, which is4 (since2*4=8≡1 mod7). So m≡4 mod7. Thus, m=7n +4. Then k=30(7n +4)=210n +120. Therefore, k≡120 mod210.

Similarly, compute all16 combinations. However, this would take time. But in any case, the solutions mod210 are16 numbers. Let's see how many of these are less than15.

The16 solutions mod210 are:

1. 0 (excluded as before)

2. 1 (since k≡1 mod2,3,5,7: k≡1 mod210)

3. Numbers like 120 (as above), etc.

等等，也许我们可以枚举所有可能的组合：

For each prime factor, choose k≡0 or1 modp. So for primes2,3,5,7:

Each prime can be either0 or1, so16 combinations.

Compute each combination:

1. All zeros: k≡0 mod2,3,5,7 =>k≡0 mod210. k=0 (excluded).

2. All ones: k≡1 mod2,3,5,7 =>k≡1 mod210. k=1 (excluded, <15).

3. 0 mod2, 0 mod3, 0 mod5, 1 mod7: k≡120 mod210 as above.

Similarly, compute others. For example:

- 0 mod2,0 mod3,1 mod5,1 mod7: Compute k≡0 mod6, k≡1 mod5, k≡1 mod7.

Compute k≡0 mod6 and k≡1 mod5 and k≡1 mod7.

First, k≡0 mod6 and k≡1 mod5. Let k=6m=5n +1. Then6m≡1 mod5 =>m≡1 mod5. So m=5p +1. Then k=6*(5p +1)=30p +6. Now, k≡1 mod7:30p +6≡1 mod7 =>30p≡-5 mod7 =>30≡2 mod7, so2p≡2 mod7 =>p≡1 mod7. Thus, p=7q +1. Then k=30*(7q +1)+6=210q +36. Therefore, k≡36 mod210.

Similarly, compute all combinations. But this is tedious. However, the key point is that the16 solutions mod210 will be numbers from0 to209. Since we need k>=15, the number of solutions in [15,210] is16 minus the number of

solutions less than15.

So how many of the16 solutions are less than15?

The solutions include:

1. k≡0 mod210:0

2. k≡1 mod210:1

3. k≡1 mod2,3,5,7:1

等等，不对。等等，k ≡ 1 mod 2,3,5,7：k ≡ 1 mod 210，因此 k = 1。

Other solutions:

For example, k≡0 mod2,3,5 and≡1 mod7: as above, k≡120 mod210. 等等，120。

Similarly, other combinations will result in numbers like36, etc.

So likely, the solutions less than15 are only0 and1, which are excluded. Therefore, all16 solutions are>=15? But16 solutions in [0,210) are0,1, and14 others between1 and209. But if0 and1 are excluded, then14 solutions in [1,210). But ceil(sqrt(211))=15, so we need k>=15. Therefore, the number of solutions is14 (since0 and1 are excluded, but some solutions might be between2 and14). Wait, but if all solutions are>=15, then we have14 solutions. However, let's check.

For example, k≡1 mod210:1 (excluded). k≡120 mod210:120. k≡36 mod210:36. Similarly, other combinations will result in numbers >=36, 120, etc. So likely, all solutions except0 and1 are>=36, which is>=15. Therefore, the number of solutions in [15,210] is16 -2=14. Therefore,14 beautiful numbers. But wait, 14 is more than ten. Therefore, b=211 would have14 beautiful numbers. Therefore, the minimal b is211. But the problem asks for the least integer b>=2 for which there are more than ten b-beautiful integers. Therefore,211. But let's confirm.

But before confirming, let's check if there are smaller b with more solutions. For example, if (b -1) is a product of three primes, say235=30, so b=31. Then the number of solutions mod30 is2^3=8. Therefore, in the interval [ceil(sqrt(31))≈6,30], the number of solutions would be8 minus those less than6. Let's compute.

Solutions to k(k -1)≡0 mod30. Which requires k≡0 or1 mod2,3,5.

Using Chinese Remainder Theorem, the solutions are combinations of:

- k≡0 or1 mod2

- k≡0 or1 mod3

等等，3 个素数：2、3、5。

So the combinations are:

1. k≡0 mod2,3,5: k≡0 mod30

2. k≡0 mod2,3 and≡1 mod5: k≡0 mod6 and≡1 mod5. Solve k=6m≡1 mod5 =>6m≡1 mod5 =>m≡1 mod5. So m=5n +1. Then k=6*(5n +1)=30n +6≡6 mod30.

3. k≡0 mod2,5 and≡1 mod3: k≡0 mod10 and≡1 mod3. Solve k=10m≡1 mod3 =>10m≡1 mod3 =>m≡1 mod3. So m=3n +1. Then k=10*(3n +1)=30n +10≡10 mod30.

等等，10 mod 3 = 1。因此，10 ≡ 1 mod 3。所以，这是正确的。

4. k≡0 mod2 and≡1 mod3,5: k≡0 mod2 and≡1 mod3,5. Solve k≡1 mod3,5 and≡0 mod2. Find k≡1 mod15 and≡0 mod2. Solve k=15m +1≡0 mod2 =>15m +1≡m +1 mod2≡0 =>m≡1 mod2. So m=2n +1. Then k=15*(2n +1)+1=30n +16≡16 mod30.

Similarly, other combinations:

5. k≡0 mod3,5 and≡1 mod2: k≡0 mod15 and≡1 mod2: k=15m≡1 mod2 =>15m≡1 *m≡1 mod2 =>m≡1 mod2. So m=2n +1. Then k=15(2n +1)=30n +15≡15 mod30.*

6. k≡0 mod3 and≡1 mod2,5: k≡0 mod3 and≡1 mod2,5. Solve k≡1 mod2,5 and≡0 mod3. k≡1 mod10 and≡0 mod3. Solve k=10m +1≡0 mod3 =>10m +1≡m +1≡0 mod3 =>m≡2 mod3. So m=3n +2. Then k=10*(3n +2)+1=30n +21≡21 mod30.

7. k≡0 mod5 and≡1 mod2,3: k≡0 mod5 and≡1 mod2,3. Solve k≡0 mod5 and≡1 mod6. k=5m≡1 mod6. 5m ≡ 1 mod 6。由于 5 ≡ -1 mod 6，因此方程变为 -m ≡ 1 mod 6，即 m ≡ 5 mod 6。所以，m = 6n + 5。接着，k = 5*(6n + 5) = 30n + 25 ≡ 25 mod 30。

8. k≡1 mod2,3,5: k≡1 mod30.

Thus, the8 solutions mod30 are0,6,10,15,16,21,25,30. 等等，0 和 30 是相同的。所以这些解为：0, 6, 10, 15, 16, 21, 25, 1（因为当 k=1 时，模 30 下的结果是 1）。等等，不对。等等，第 8 个解是 $k \equiv 1 \mod 30$。所以这些解是 $k \equiv 0, 1, 6, 10, 15, 16, 21, 25 \mod 30$。

But in the interval [6,30], the solutions are6,10,15,16,21,25,30≡0 mod30 (excluded), and1≡1 mod30 (excluded). So valid k=6,10,15,16,21,25.

Check each:

k=6:

(36 -6)/30=30/30=1. d1=1, d2=6 -1=5. n=36. In base31,36=1*31 +5=36. Digits1 and5 sum to6. Correct.

k=10:

(100 -10)/30=90/30=3. d1=3, d2=10 -3=7. n=100. In base31,100=3*31 +7=93 +7=100. Digits3 and7 sum to10. Correct.

k=15:

(225 -15)/30=210/30=7. d1=7, d2=15 -7=8. n=225. In base31,225=7*31 +8=217 +8=225. Digits7 and8 sum to15. Correct.

k=16:

(256 -16)/30=240/30=8. d1=8, d2=16 -8=8. n=256. In base31,256=8*31 +8=248 +8=256. Digits8 and8 sum to16. Correct.

k=21:

(441 -21)/30=420/30=14. d1=14, d2=21 -14=7. n=441. In base31,441=14*31 +7=434 +7=441. Digits14 and7 sum to21. Correct.

等等，21² = 441。是的。

k=25:

(625 -25)/30=600/30=20. d1=20, d2=25 -20=5. n=625. In base31,625=20*31 +5=620 +5=625. Digits20 and5 sum to25. Correct.

Thus, six beautiful numbers:36,100,225,256,441,625. So six solutions. Which is more than the previous examples, but still less than ten. Therefore, b=31 has six beautiful numbers.

Therefore, moving to b=211 would give14, which is more than ten. But before that, check if there is a smaller b. For example, take (b -1)=2·3·5=30, leading to b=31 with six solutions. If we take (b -1)=2·3·5·7=210, leading to b=211 with14 solutions. Alternatively, maybe a smaller (b -1) with three prime factors, but with overlapping solutions.

Alternatively, take (b -1)=2·3·7=42, so b=43. Then number of solutions mod42 is2^3=8. Therefore, in the interval [ceil(sqrt(43))≈7,42], the number of solutions would be8 minus those less than7.

Compute solutions mod42:

Solutions are combinations of k≡0 or1 mod2,3,7.

Therefore,8 solutions:

1.  k≡0 mod2,3,7: k≡0 mod42.

等待：

1.  0 mod2,3,7:0 mod42.

等等，这样会得到以下 8 种组合：

For each prime, choose 0 or1:

- 0,0,0:0 mod42.

等等，这会引发一些疑惑。我们来枚举所有组合吧：

1.  0 mod2,0 mod3,0 mod7:0 mod42.

等等，这里可能需要同时满足以下条件：k mod 2 = 0，k mod 3 = 0，k mod 7 = 0。因此，k ≡ 0 mod 42。

2.  0 mod2,0 mod3,1 mod7:0 mod6,1 mod7. Solve k=6m≡1 mod7. 6m≡1 mod7. 6≡-1 mod7, so -m≡1 mod7 =>m≡-1≡6 mod7. So m=7n +6. Then k=6*(7n +6)=42n +36≡36 mod42.

3.  0 mod2,1 mod3,0 mod7:0 mod2,0 mod7. So k≡0 mod14. Also,1 mod3. Solve k=14m≡1 mod3. 14≡2 mod3, so2m≡1 mod3 =>m≡2 mod3. So m=3n +2. Then k=14*(3n +2)=42n +28≡28 mod42.

等等，28 mod 3 = 1，因为 28 ÷ 3 = 9 ... 1。所以，28 ≡ 1 mod 3。这是正确的。

4. 0 mod2,1 mod3,1 mod7:0 mod2,1 mod3,1 mod7. Solve k≡1 mod3,1 mod7, and0 mod2. k≡1 mod21 and0 mod2. Since21 is odd, k≡1 mod21 and0 mod2. But1≡1 mod21 and0 mod2, which is impossible. Therefore, no solution here.

等等，这有点让人困惑。我们来仔细看看。

k≡1 mod3 and1 mod7: k≡1 mod21.

k≡0 mod2.

So k≡1 mod21 and0 mod2. So k must be even and≡1 mod21. But1 mod21 is odd. Therefore, no solution.

Therefore, this combination is invalid.

So combination4 is invalid.

5. 1 mod2,0 mod3,0 mod7:1 mod2,0 mod21. Similarly, impossible. Because0 mod21 is even, but1 mod2 is odd. Therefore, no solution.

6. 1 mod2,0 mod3,1 mod7:1 mod2,0 mod3,1 mod7. Solve k≡1 mod2,0 mod3,1 mod7.

k≡1 mod2 and0 mod3. Let k=3m≡1 mod2. 3m≡1 mod2 =>m≡1 mod2. So m=2n +1. Then k=3*(2n +1)=6n +3≡3 mod6.

Also, k≡1 mod7. So6n +3≡1 mod7 =>6n≡-2≡5 mod7. Multiply both sides by inverse of6 mod7, which is6 (since6*6=36≡1 mod7). So n≡5*6=30≡30 -4*7=2 mod7. So n=7p +2. Then k=6(7p +2)+3=42p +15. Therefore, k≡15 mod42.

等等，15 mod 2 = 1，这是正确的，因为 15 是奇数。

k=15≡0 mod3,3*5.

k=15≡1 mod7, since15=2*7 +1.

Thus, valid.

7. 1 mod2,1 mod3,0 mod7:1 mod2,1 mod3,0 mod7. Solve k≡1 mod2,1 mod3,0 mod7.

k≡1 mod2 and1 mod3. Then k≡4 mod6.

Also, k≡0 mod7. Solve k=7m≡4 mod6. 7m≡m≡4 mod6. So m≡4 mod6. m=6n +4. Then k=7*(6n +4)=42n +28≡28 mod42.

Check k=28:

28 mod2=0≡1 mod2? No. Wait, 28 is even, but we need k≡1 mod2. So invalid. Therefore, no solution here.

等等，这里有问题。我们重新解决一下：

k≡1 mod2,1 mod3,0 mod7.

So k=7m, and k≡1 mod2 and1 mod3.

7m≡1 mod2 => m≡1 mod2.

7m≡1 mod3 => 7m≡m≡1 mod3.

Therefore, m≡1 mod2 and m≡1 mod3. So m≡1 mod lcm(2,3)=6. Thus, m=6n +1. Then k=7*(6n +1)=42n +7≡7 mod42.

Therefore, k≡7 mod42. Check:

7 mod2=1, 7 mod3=1, 7 mod7=0. Correct. So k=7. But7 is less than ceil(sqrt(43))≈6.93, so ceil is7. So k=7 is included. Therefore, valid.

8.  1 mod2,1 mod3,1 mod7:1 mod2,1 mod3,1 mod7. Solve k≡1 mod2,1 mod3,1 mod7. Which is k≡1 mod lcm(2,3,7)=42. So k≡1 mod42.

Thus, solutions are k≡0,7,15,21,28,35,42≡0 mod42, but k=7,15,21,28,35,1.

But in the interval [7,42], the solutions are7,15,21,28,35,42. However,42 is equivalent to0 mod42, which is excluded. So valid k=7,15,21,28,35.

Check each:

k=7:

(49 -7)/42=42/42=1. d1=1, d2=7 -1=6. n=49. In base43,49=1*43 +6=49. Digits1 and6 sum to7. Correct.

k=15:

(225 -15)/42=210/42=5. d1=5, d2=15 -5=10. n=225. In base43,225=5*43 +10=215 +10=225. Digits5 and10 sum to15. Correct.

等等，5 + 10 = 15。没错。

k=21:

(441 -21)/42=420/42=10. d1=10, d2=21 -10=11. n=441. In base43,441=10*43 +11=430 +11=441. Digits10 and11 sum to21. Correct.

k=28:

(784 -28)/42=756/42=18. d1=18, d2=28 -18=10. n=784. In base43,784=18*43 +10=774 +10=784. Digits18 and10 sum to28. Correct.

k=35:

(1225 -35)/42=1190/42=28.5. Not integer. 等等，1190 除以 42 是 28.333...。等等，这不可能。

等等，$1225 - 35 = 1190$。$1190$ 除以 $42$：$42 \times 28 = 1176$，$1190 - 1176 = 14$。所以 $1190 = 42 \times 28 + 14$。因此，$1190/42 = 28 + 14/42 = 28 + 1/3$。这不是整数。因此，$k = 35$ 不是有效解。等等，但在模 $42$ 下，$k = 35$ 是一个解。然而，$(k^2 - k)/42$ 并不是整数。这是因为虽然 $42$ 可以整除 $k(k-1)$，但 $k = 35$ 时，$k \equiv 35 \pmod{42}$，$k - 1 = 34 \pmod{42}$。$35$ 和 $34$ 在模 $42$ 下分别是 $35$ 和 $34$，但它们的乘积是 $35 \times 34 = 1190$。$1190$ 模 $42$ 等于 $14$，因为 $1190 \div 42 = 28$ 余 $14$。

因此，$1190 \equiv 14 \pmod{42}$。也就是说，42 可以整除 1190 除以 42 的结果，但 1190/42 并不是整数。这表明虽然满足模 42 的条件，但 $(k^2 - k)/42$ 并不一定是整数。等等，但 $k(k-1)$ 必须能被 42 整除。然而，$35 \cdot 34 = 1190$，而 1190 并不能被 42 整除。等等，42 是 $2 \cdot 3 \cdot 7$。接下来检查：

1190 divided by2:1190/2=595. Divided by3:595/3≈198.333, not integer. Divided by7:1190/7=170. So 1190=$2 \cdot 5 \cdot 7 \cdot 17$. *Therefore, 42=$2 \cdot 3 \cdot 7$ divides1190? 42=$2 \cdot 3 \cdot 7$, but1190=$2 \cdot 5 \cdot 7 \cdot 17$. The 3 is missing. Therefore,42 does not divide1190.* Therefore, k=35 is not a valid solution. Wait, but according to the previous analysis, k=35 is a solution mod42. But actually, k=35≡-7 mod42, but k(k-1)=$35 \cdot 34$=1190. 1190 *mod42=1190* -$42 \cdot 28$=1190 -1176=14. Therefore,14≡0 mod14, but42 does not divide1190. Therefore, there is a contradiction here. Therefore, our earlier assumption that combining the modular solutions would lead to k(k -1)≡0 mod42 is incorrect. Because even though k≡0 or1 mod each prime, the product k(k -1) might not be divisible by the product of the primes if the factors are distributed across the primes.

等等，这到底是怎么回事？让我们来仔细思考一下。假设我们需要满足 $k(k-1) \equiv 0 \mod 42$。由于 $42 = 2 \cdot 3 \cdot 7$，因此 $2 \cdot 3 \cdot 7$ 都必须整除 $k(k-1)$。因为 $k$ 和 $k-1$ 是互质的，所以每个素数必须整除 $k$ 或 $k-1$。也就是说，对于每个素数 $p$（这里为 $2, 3, 7$），必须满足 $p$ 整除 $k$ 或 $p$ 整除 $k-1$。因此，对于每个素数，我们有 $k \equiv 0 \mod p$ 或 $k \equiv 1 \mod p$。因此，该同余方程的解是所有可能的组合，即每个素数取 0 或 1 的情况。对于三个素数，共有 $2^3 = 8$ 种组合。

然而，当我们取 $k = 35$ 时，它在模 2、3、7 下分别满足 $k \equiv 1 \mod 2$、$k \equiv 2 \mod 3$、$k \equiv 0 \mod 7$。等等，不对。等等，$35 \mod 2 = 1$，$35 \mod 3 = 2$，$35 \mod 7 = 0$。再等等，根据 3 来说，$35 \equiv 2 \mod 3$，这并不是 0 或 1 $\mod 3$。因此，$k = 35$ 并不满足 $k \equiv 0$ 或 $1 \mod 3$ 的条件。等等，但在之前的分析中，我们考虑的是 $k \equiv 0$ 或 $1 \mod 2$、0 或 1 $\mod 3$、0 或 1 $\mod 7$ 的组合。对于 $k = 35$，我们有 $k \equiv 1 \mod 2$、$k \equiv 2 \mod 3$、$k \equiv 0 \mod 7$。但这并不是一个合法的组合。等等，这说明哪里出错了？

等等，不对。当考虑模 42 的解时，解的集合是所有满足以下条件的 $k$：对每个素因子 $p$，都有 $k \equiv 0 \pmod{p}$ 或 $k \equiv 1 \pmod{p}$。对于 $p = 2$，$k \equiv 1 \pmod{2}$；对于 $p = 3$，$k \equiv 2 \pmod{3}$；对于 $p = 7$，$k \equiv 0 \pmod{7}$。然而，$2 \pmod 3$ 既不是 0 也不是 1 $\pmod 3$，所以 $k = 35$ 并不满足条件。因此，之前的分析中存在错误。

在我们之前的分析中，我们寻找了满足以下条件的解：
对于每个素因子 $p$，都有 $k \equiv 0 \pmod{p}$ 或 $1 \pmod{p}$。
因此，$k = 35$ 对于 $p = 7$ 满足 $0 \pmod 7$，但对 $p = 3$ 满足 $2 \pmod 3$，这不符合条件。所以，$k = 35$ 并不算是一个解。

然而，在我们列举的 8 个解中，$k = 35$ 被错误地包含进去了。让我们重新审视一下之前的解法。

We had combinations:

1. 0,0,0:0 mod42.

2. 0,0,1:36 mod42.

3. 0,1,0:28 mod42.

4. Invalid.

等等，第四种组合是无效的。

5. Invalid.

6.15 mod42.

7.7 mod42.

等等，第七种是 $k \equiv 7 \pmod{42}$ 。

8.1 mod42.

But in the interval [7,42], the valid solutions are7,15,21,28,35. But35 is invalid as above. Similarly,21:

k=21:

Check k=21. For p=2:21≡1 mod2. For p=3:21≡0 mod3. For p=7:21≡0 mod7. Therefore, for p=3 and p=7,21≡0 mod3 and0 mod7, which is allowed (since0 is allowed). So k=21 is a valid solution. Because for each prime, k≡0 mod p or1 mod p. For p=2,21≡1; for p=3,21≡0; for p=7,21≡0. Therefore, valid.

But k=21:

(441 -21)/42=420/42=10. Which is integer. So valid. Therefore, n=441=21². In base43,441=10*43 +11=430 +11=441. Digits10 and11 sum to21. Correct.

Similarly, k=28:

k=28:

For p=2:28≡0 mod2.

p=3:28≡1 mod3.

p=7:28≡0 mod7.

Therefore, for p=2,28≡0; p=3,28≡1; p=7,28≡0. Therefore, valid.

Thus, k=28 is valid. But earlier, we thought k=28 would lead to non-integer, but actually:

(784 -28)/42=756/42=18. Which is integer. Therefore, d1=18, d2=28 -18=10. n=784=28². In base43,784=18*43 +10=774 +10=784. Digits18 and10 sum to28. Correct.

Therefore, the valid k's in [7,42] are7,15,21,28. k=35 is invalid. So four solutions. Therefore, in base43, there are four beautiful numbers. Which is more than previous examples but still less than ten.

Therefore, moving to b=211 with14 solutions is the first instance where the number exceeds ten. Therefore, the minimal b is211.

But let's check another example. Let's take (b -1)=2*3*5*7=210, b=211. Then the number of solutions is16. As before, in the interval [15,210], assuming all solutions are>=15, we have16 solutions. Each solution corresponds to a combination of k≡0 or1 mod2,3,5,7. For example:

k≡0 mod2,3,5,7: k=0 mod210. Excluded.

k≡1 mod2,3,5,7: k=1 mod210. Excluded.

Then, other combinations like:

k≡0 mod2,3,5 and1 mod7: k=120 mod210 as before.

Similarly, each combination gives a unique k in [0,210). The solutions are:

0,1,30,36,70,84,90,120,126,150,156,180,186,210≡0. So the solutions are0,1,30,36,70,84,90,120,126,150,156,180,186. But0 and1 are excluded. Now check which of these are>=15.

Starting from30 upwards:

30,36,70,84,90,120,126,150,156,180,186. So eleven solutions. But wait,120,126,150,156,180,186. Let's count:

From30 to186, how many solutions? Let's list them:

30,36,70,84,90,120,126,150,156,180,186. That's eleven numbers. Wait, but earlier I thought there were16 solutions. But perhaps some solutions are less than15. For example, k=6 mod70? Wait, no. Let me recount.

The solutions mod210 are:

For each prime, choose0 or1. So16 combinations. The solutions are:

1. 0 mod210.

2.1 mod210.

3.0 mod2,3,5 and1 mod7:30*1=30 mod210.

等等，不对：

Wait, for k≡0 mod2,3,5 and1 mod7. As before, k=30m≡1 mod7. 30 除以 7 的余数为 2，因此 $30m \equiv 2m \equiv 1 \pmod{7}$。求解 $m$：

$2m \equiv 1 \pmod{7}$，其解为 $m \equiv 4 \pmod{7}$，因为 $2 \times 4 = 8 \equiv 1 \pmod{7}$。

于是，$m = 7n + 4$，代入可得：

$k = 30 \times (7n + 4) = 210n + 120$。

因此，$k \equiv 120 \pmod{210}$。

Similarly, other combinations:

k≡0 mod2,3,7 and1 mod5: Solve k≡0 mod42 and1 mod5. k=42m≡1 mod5. 42 ≡ 2 mod 5，因此 2m ≡ 1 mod 5。解得 m ≡ 3 mod 5。所以 m = 5n + 3，代入得：

k = 42 × (5n + 3) = 210n + 126 ≡ 126 mod 210。

Similarly, others:

k≡0 mod2,5,7 and1 mod3: Solve k≡0 mod70 and1 mod3. 70m≡1 mod3 =>70≡1 mod3, so1*m≡1 mod3*=>*m≡1 mod3. m=3n +1. Then k=70*(3n +1)=210n +70≡70 mod210.

Thus, solutions like70,126,120, etc.

Listing all16 solutions mod210:

0,1,30,36,70,84,90,120,126,150,156,180,186,210 (which is0), etc. Wait, actually, the solutions are all numbers where for each prime, they are≡0 or1. Therefore, the numbers are:

For each prime factor2,3,5,7:

- If we choose0 for all, we get0.

- If we choose1 for all, we get1.

- Other combinations:

For example:

- 0 mod2,3,5,1 mod7:120

- 0 mod2,3,7,1 mod5:126

等等，我需要重新生成这16个解。

But to save time, let's note that in the interval [15,210], the solutions are from30 up to186, which are eleven numbers. Wait, but earlier we thought there were16 solutions. Therefore, perhaps my previous assumption that all solutions are>=15 is incorrect. Let's check some solutions:

For example, k≡0 mod2,3,5,1 mod7:120

k≡0 mod2,3,7,1 mod5:126

k≡0 mod2,5,7,1 mod3:70

k≡0 mod3,5,7,1 mod2:105 (Wait, 105 mod2=1, but105 is odd. Let's check:

If we choose k≡1 mod2,0 mod3,5,7. Then k=0 mod105 and1 mod2. Solve k=105m≡1 mod2. 105≡1 mod2, so m≡1 mod2. m=2n +1. Then k=105*(2n +1)=210n +105≡105 mod210. So k=105. Check105:

105 in base211 is a two-digit number. Let's check if it's b-beautiful.

n=105^2=11025. In base211,11025 divided by211:

*2115*2=10972, 11025-10972=53. So11025=52*211 +53. Digits52 and53. Their sum is52+53=105. sqrt(11025)=105. Correct.

Therefore, k=105 is a valid solution. Similarly, k=105 is in the interval [15,210].

Similarly, k=165:

Another combination. For example, k≡1 mod2,0 mod3,5,7. Then k=0 mod105 and1 mod2. As above, k=105m +105. Wait, no.

等等，105 mod 2 = 1，所以 k=105 ≡ 1 mod 2。但如果选择条件是 k ≡ 0 mod 3,5,7 和 k ≡ 1 mod 2，那么 k=105 是符合条件的。

Similarly, k=105 is a solution. So in total, the solutions in [15,210] are:

30,36,70,84,90,105,120,126,150,156,180,186. Let's count these:

From30 to186, how many numbers?

30,36 (difference6),70 (difference34),84 (difference14),90 (difference6),105 (difference15),120 (difference15),126 (difference6),150 (difference24),156 (difference6),180 (difference24),186 (difference6). Total of11 numbers. Wait, but this is only11. But we need16 solutions. Where are the other5?

等等，可能还有其他的解。例如，k ≡ 0 mod 2,3,1 mod 5,7。

Let's compute:

k≡0 mod6 and1 mod5,1 mod7.

Solve k≡0 mod6 and k≡1 mod5,1 mod7.

First, solve k≡0 mod6 and k≡1 mod5.

k=6m≡1 mod5 =>6m≡1 mod5 =>m≡1 mod5. m=5n +1. Then k=6*(5n +1)=30n +6≡6 mod30.

Then solve k≡6 mod30 and k≡1 mod7.

k=30n +6≡1 mod7 =>30n +6≡1 mod7 =>30≡2 mod7, so2n +6≡1 mod7 =>2n≡-5≡2 mod7 =>n≡1 mod7. n=7p +1. Then k=30*(7p +1)+6=210p +36. Therefore, k≡36 mod210.

Therefore, k=36. Check k=36:

$(36^2 -36)/210=(1260)/210=6$. d1=6, d2=36 -6=30. n=1296. In base211,1296=6*211 +30=1266 +30=1296. Digits6 and30 sum to36. Correct.

Thus, k=36 is a solution. So this was missing earlier. Therefore, the solutions include numbers like36,70,105, etc. Therefore, the total number of solutions in [15,210] is16 minus the solutions less than15. The solutions less than15 are0 and1. Therefore,16 -2=14 solutions. Therefore,14 solutions. Each of these k's from30 to210 (but actually, some are less than15? Wait, no. k=36 is in the interval [15,210]. Similarly, k=30,36,70,84,90,105,120,126,150,156,180,186. That's twelve numbers. Wait, but16 -2=14. So where are the other two?

Wait, perhaps k=165 and k=168.

等等，让我检查一下解法。

Another combination is k≡0 mod2,5,1 mod3,7.

Solve k≡0 mod10 and1 mod3,1 mod7.

k≡0 mod10 and1 mod3: k=10m≡1 mod3 =>10m≡1 mod3 =>m≡1 mod3. m=3n +1. k=10*(3n +1)=30n +10≡10 mod30.

Then solve k≡10 mod30 and1 mod7.

k=30n +10≡1 mod7 =>30n +10≡1 mod7 =>30≡2 mod7, so2n +10≡1 mod7 =>2n≡-9≡-2≡5 mod7 =>n≡(5)*(inverse of2 mod7)=54=20≡6 mod7. n=7p +6. Then k=30*(7p +6)+10=210p +190. Thus, k≡190 mod210.

Check k=190:

$(190^2 -190)/210=(36100 -190)/210=35910/210=171$. d1=171, d2=190 -171=19. n=36100. In base211, 36100=171*211 +19=36081 +19=36100. Digits171 and19 sum to190. Correct.

Thus, k=190 is a solution. Similarly, k=190 is in the interval.

Similarly, another combination:

k≡0 mod2,7,1 mod3,5.

Solve k≡0 mod14 and1 mod3,1 mod5.

k≡0 mod14 and1 mod3:14m≡1 mod3 =>14≡2 mod3, so2m≡1 mod3 =>m≡2 mod3. m=3n +2. k=14*(3n +2)=42n +28≡28 mod42.

Then solve k≡28 mod42 and1 mod5.

k=42n +28≡1 mod5 =>42n +28≡1 mod5 =>42≡2 mod5, 28≡3 mod5. So 2n +3≡1 mod5 =>2n≡-2≡3 mod5 =>n≡(3)*(inverse of2)=33=9≡4 mod5. n=5p +4. Then k=42*(5p +4)+28=210p +176 +28=210p +204. Wait, 42*4=168, 168 +28=196. Wait, maybe miscalculation.

等等，k = 42n + 28 ≡ 1 mod 5。

Compute 42n +28 mod5:

42≡2 mod5, 28≡3 mod5. So equation:2n +3≡1 mod5 =>2n≡-2≡3 mod5 =>n≡(3)*(inverse of2)=33=9≡4 mod5. Therefore, n=5p +4. Then k=42*(5p +4)+28=210p +168 +28=210p +196≡196 mod210.

Check k=196:

$(196^2 -196)/210=(38416 -196)/210=38220/210=182$. d1=182, d2=196 -182=14. n=38416. In base211, 38416=182*211 +14=38302 +14=38416. Digits182 and14 sum to196. Correct.

Thus, k=196 is a solution.

Therefore, the solutions in [15,210] are:

30,36,70,84,90,105,120,126,150,156,180,186,190,196. That's14 solutions. Therefore,14 solutions. Each corresponding to a combination of choosing0 or1 mod each prime. Therefore, each solution is unique, and they are all>=15. Therefore, in base211, there are14 beautiful numbers, which is more than ten. Therefore, the minimal b is211.

But let's check if there is a smaller b with more solutions. For example, take (b -1)=2·35*11=330, so b=331. Then the number of solutions would be2^4=16. However, ceil(sqrt(331))≈18.3, so interval[19,330]. The solutions mod330 would be16 numbers. If all are>=19, then16 solutions. But likely, some solutions are less than19, so the count would be16 minus those less than19. But this might result in more than14 solutions. However, this would require checking.

Alternatively, take (b -1)=2·3·5·7=210, b=211, which gives14 solutions. The next candidate would be (b -1)=2·3·5·7*11=2310, b=2311, giving32 solutions. But this is much larger.

Alternatively, check b=105. Then (b -1)=104=8·13. The number of solutions would be related to the factors of8 and13. Since8=2^3 and13 is prime. The number of solutions mod8 isφ(8)=4? Wait, no. For modulus8, the number of solutions to k(k -1)≡0 mod8 is2 (as k≡0 or1 mod8). Similarly, for modulus13, the number of solutions is2. Therefore, total solutions mod104=8·13 is2*2=4. Therefore, in the interval [ceil(sqrt(105))≈11,104], the number of solutions is4. Thus, four solutions. Which is less than ten.

Therefore, the minimal b is211.

**Final Answer**

\boxed{211}

<|end_of_thought|>

<|begin_of_solution|>

To determine the least integer $b \geq 2$ for which there are more than ten $b$-beautiful integers, we need to find a base $b$ where there are more than ten two-digit numbers $n$ in base $b$ such that the sum of their digits equals $\sqrt{n}$.

A number $n$ is $b$-beautiful if:

1. $n$ is a two-digit number in base $b$, i.e., $b \leq n \leq b^2 - 1$.
2. The sum of the digits of $n$ in base $b$ equals $\sqrt{n}$.

Key steps:

1. For $n$ to be a two-digit number in base $b$, $n$ must be a perfect square $k^2$ where $k$ is an integer such that $\lceil \sqrt{b} \rceil \leq k \leq b - 1$.
2. The digits of $k^2$ in base $b$ must sum to $k$. This translates to the condition that $(k^2 - k)$ is divisible by $(b - 1)$.

To find the smallest $b$ such that there are more than ten such $k$, we need $(b - 1)$ to have many divisors. The number of solutions is related to the number of integers $k$ such that $(b - 1)$ divides $k(k - 1)$.

By analyzing the problem, we find that choosing $b - 1$ as the product of the first few primes (square-free) maximizes the number of solutions. The smallest such $b$ is found when $b - 1 = 2 \times 3 \times 5 \times 7 = 210$, leading to $b = 211$.

In this case, the number of solutions (beautiful numbers) is determined by the combinations of residues modulo the prime factors of 210. This results in 16 solutions modulo 210, all of which are valid in the range \( \lceil \sqrt{211} \rceil = 15 \) to 210, giving us 14 valid solutions (more than ten).

Thus, the least integer \( b \geq 2 \) for which there are more than ten \( b \)-beautiful integers is \(\boxed{211}\).

<|end_of_solution|>

GT: 211

Predict: 211 ✅