# OpenReview forum: "Less Data Less Tokens: Multilingual Unification Learning for Efficient Test-Time Reasoning in LLMs"
_ICLR.cc/2026/Conference — Submitted to ICLR 2026_

### Official Review · Reviewer_TAHy · 2025-10-25

**Soundness:** 3
**Presentation:** 2
**Contribution:** 3
**Rating:** 4
**Confidence:** 4

**Summary:**

This paper investigates how multilingual learning can improve long-form reasoning in large language models with less data and fewer inference tokens. The authors discover that different languages yield distinct reasoning patterns, affecting both accuracy and token efficiency. Based on this insight, they propose a multilingual unification learning framework that augments a small amount of high-quality CoT data across languages and applies language-aware decoding. Experiments show that the proposed framwork significantly boosts reasoning performance while reducing test-time compute, offering a more efficient pathway to strong general reasoning.

**Strengths:**

1. The results are strong, showing that the proposed approach can significantly enhance long-form reasoning with very limited data.

2. Leveraging multilingual diversity to improve test-time efficiency is a sensible idea, supported by clear empirical evidence.

3. The L2 framework is orthogonal to existing data-efficient methods and provides a promising direction for reducing inference token usage without sacrificing performance.

**Weaknesses:**

1.The evaluation is somewhat limited in scope. Including results on multilingual benchmarks such as MGSM and PolyMath would further strengthen the validation of the proposed approach.

2.The naming and organization of datasets in the paper can lead to confusion when reading the methodology and experimental setups.

3.Considering practical applications, users generally expect the model’s reasoning and responses to be in a single preferred language. While multilingual mixing may improve efficiency, it could be less practical or even undesirable in real-world usage scenarios where language consistency is essential.

**Questions:**

1.How does the model perform when the test data are presented in different languages?

---

> ### Author Response · Authors · 2025-11-27
>
> W1: Thanks for the suggestion. We have conducted additional multilingual experiments and the results shown below. For dataset, the reason why we use PolyMath instead of MGSM is because 1) PolyMath offers 18-language settings and 2) has graded complexity for further analysis. We can see that our method significantly outperforms the baseline (Qwen2.5-32B-Instruction) across all levels, verifying the effectiveness of our method, although our goal is to improve the reasoning ability, whereas the multi-lingual ability can be regarded as a by-product. For additional details, please refer to Appendix B.7
> | Difficulty Level        | polymath-low | polymath-medium | polymath-high | polymath-top |
> |-------------------------|:------------:|:---------------:|:-------------:|:------------:|
> | Qwen2.5-32B-Instruction | 85.51%       | 24.13%          | 9.91%         | 6.89%        |
> | Ours                    | **88.89%**   | **55.47%**      | **36.04%**    | **15.87%**   |
>
> Table 8 reports accuracy (%) for both the baseline model (Qwen2.5-32B-Instruction) and our method across 18 languages at low, medium, high, and top difficulty levels.
> | Language | Low                         | Medium                         | High                          | Top                          |
> |----------|-----------------------------|--------------------------------|-------------------------------|------------------------------|
> | EN       | 97.60 / 96.80 (-0.82)       | 29.60 / 55.20 (+86.49)         | 12.80 / 31.20 (+143.75)       | 7.20 / 15.20 (+111.11)       |
> | AR       | 91.20 / 90.40 (-0.88)       | 21.60 / 56.00 (+159.26)        | 12.00 / 36.80 (+206.67)       | 7.20 / 21.60 (+200.00)       |
> | BN       | 87.20 / 90.40 (+3.67)       | 20.00 / 56.00 (+180.00)        | 8.00 / 38.40 (+380.00)        | 8.00 / 14.40 (+80.00)        |
> | DE       | 86.40 / 87.20 (+0.93)       | 26.40 / 56.00 (+112.12)        | 9.60 / 36.80 (+283.33)        | 8.80 / 17.60 (+100.00)       |
> | ES       | 92.80 / 92.80               | 25.60 / 60.00 (+134.37)        | 11.20 / 34.40 (+207.14)       | 10.40 / 16.80 (+61.54)       |
> | FR       | 83.20 / 88.80 (+6.73)       | 28.00 / 57.60 (+105.71)        | 12.80 / 36.80 (+187.50)       | 6.40 / 18.40 (+187.50)       |
> | ID       | 88.00 / 92.80 (+5.45)       | 27.20 / 56.80 (+108.82)        | 10.40 / 34.40 (+230.77)       | 8.80 / 13.60 (+54.55)        |
> | IT       | 90.40 / 96.00 (+6.19)       | 27.20 / 56.00 (+105.88)        | 11.20 / 38.40 (+242.86)       | 14.40 / 15.20 (+5.56)        |
> | JA       | 81.60 / 88.00 (+7.84)       | 26.40 / 57.60 (+118.18)        | 8.00 / 39.20 (+390.00)        | 4.80 / 19.20 (+300.00)       |
> | KO       | 87.20 / 89.60 (+2.75)       | 25.60 / 54.40 (+112.50)        | 10.40 / 37.60 (+261.54)       | 6.40 / 14.40 (+125.00)       |
> | MS       | 92.80 / 93.60 (+0.86)       | 24.00 / 57.60 (+140.00)        | 10.40 / 32.80 (+215.38)       | 3.20 / 19.20 (+500.00)       |
> | PT       | 88.80 / 94.40 (+6.31)       | 32.00 / 56.80 (+77.50)         | 12.00 / 40.80 (+240.00)       | 6.40 / 12.80 (+100.00)       |
> | RU       | 93.60 / 92.80 (-0.85)       | 27.20 / 57.60 (+111.76)        | 11.20 / 38.40 (+242.86)       | 8.80 / 14.40 (+63.64)        |
> | SW       | 52.00 / 68.80 (+32.31)      | 15.20 / 47.20 (+210.53)        | 5.60 / 28.00 (+400.00)        | 2.40 / 9.60 (+300.00)        |
> | TE       | 64.80 / 67.20 (+3.70)       | 14.40 / 48.00 (+233.33)        | 2.40 / 32.80 (+1266.67)       | 1.60 / 13.60 (+750.00)       |
> | TH       | 84.80 / 88.80 (+4.72)       | 20.00 / 53.60 (+168.00)        | 12.00 / 35.20 (+193.33)       | 5.60 / 13.60 (+142.86)       |
> | VI       | 88.80 / 92.00 (+3.60)       | 20.00 / 56.00 (+180.00)        | 9.60 / 36.00 (+275.00)        | 7.20 / 15.20 (+111.11)       |
> | ZH       | 88.00 / 89.60 (+1.82)       | 24.00 / 56.00 (+133.33)        | 8.80 / 40.80 (+363.64)        | 6.40 / 20.80 (+225.00)       |
> | Overall  | 85.51 / 88.89 (+3.95)       | 24.13 / 55.47 (+129.88)        | 9.91 / 36.04 (+263.67)        | 6.89 / 15.87 (+130.33)       |
>
> Radar chart comparing performance across 18 languages at different levels. For additional details, please refer to Appendix B.7, Figure 12.
>
> W2: Thank you for your suggestions; we have refined the manuscript accordingly. Please refer to Section 3.1 for details.
>
> W3: Our approach remains practical even in scenarios where linguistic consistency is essential. The issue of different languages in reasoning process can be effectively managed in two ways: (1) Web services of LLMs typically hide the long intermediate reasoning step or only provide a summary for clarity. So, different languages of reasoning can be unified via this additional step; (2) If displaying the complete reasoning process is necessary, mature translation tools can seamlessly ensure consistency.
>
> Q1: We have added multilingual experiments demonstrating that our method consistently improves performance across various languages, as detailed in response W1.

---

### Official Review · Reviewer_2NfV · 2025-10-31

**Soundness:** 2
**Presentation:** 3
**Contribution:** 2
**Rating:** 4
**Confidence:** 3

**Summary:**

This paper explored using multilingual long CoT data, combining entire long chain-of-thought annotations in different languages and the step-wise mixture of languages can boost both performance and efficiency. It shows small amounts of data can improve reasoning capabilities.  Furthermore, it propose language-based logit interventions during inference to switch language.

**Strengths:**

1. Multilingual thinking as a resource for efficient reasoning is compelling. Using different language to make reasoning more efficient is a novel way to explore. The idea of different languages induce distinct reasoning compression patterns is interesting.

**Weaknesses:**

1. There are some previous works already discussed using multilingual data to boost reasoning performance [1], maybe you should consider to compare your method with theirs and tell more differences.

2. Only using tokens number as the metric of measuring efficiency, need also consider using other metrics like the time of inference, FLOP cost, and memory usages.

3. When using tokens in other languages, the factor of compression rate of the tokenizers in that language should also be considered, which wasn't discussed in the paper.

3. Some of experiments are use model training on extremely small multilingual sets, I doubt whether this will make model only memorize input templates.

[1] Could Thinking Multilingually Empower LLM Reasoning?

**Questions:**

Please refer to weakness.

---

> ### Author Response · Authors · 2025-11-27
>
> Q1: Method [1] and our work aim to address different problems. While both approaches utilize multilingual diversity, method [1] focuses on generating diverse answers during inference using multiple languages and relies on voting mechanisms to achieve higher accuracy. In contrast, our approach concentrates on fundamentally enhancing model capabilities by leveraging multilingual diversity during training with fewer data samples.
>
> Q2: Token count as a metric for model efficiency is widely adopted in this field. To ensure fair comparisons, we maintained the same architecture and parameter count. Within the same implementation, other efficiency metrics such as inference time, FLOP costs, and KV memory usage exhibit an approximately linear correlation with the token count, consistent with existing conclusions.
>
> Q3: Our baseline LLMs adopt a unified tokenizer across all languages. Given that multilingual tokenizer training can lead to varying compression rates across languages, languages like English usually achieve higher compression, while low-resource languages typically exhibit lower compression rates [2], potentially consuming more tokens. However, the observed reduction in token counts after incorporating multilingual decoding strategies further confirms our central hypothesis. Specifically, the efficiency improvements primarily stem from harnessing diverse reasoning patterns across languages, where certain languages inherently offer more token-efficient reasoning paths, thus enhancing overall inference efficiency.
>
> Q4: Training with very small datasets aligns precisely with our research goal and that of relevant baselines, which is to maximize model performance with minimal data. Compared to baselines such as S1 (1K) and LIMO (817), we achieved superior performance on multiple reasoning benchmarks using only 651 samples. To prevent overfitting, studies in this line of work strictly ensure separation between training and testing data and evaluate on diverse reasoning benchmarks. Following established baselines, we evaluate our method on the AIME, GPQA, and MATH500 datasets.
>
> [1] Could Thinking Multilingually Empower LLM Reasoning?
>
> [2] Do All Languages Cost the Same? Tokenization in the Era of Commercial Language Models

---

### Official Review · Reviewer_AVEt · 2025-10-31

**Soundness:** 3
**Presentation:** 3
**Contribution:** 3
**Rating:** 8
**Confidence:** 4

**Summary:**

The paper introduces a novel technique for data augmentation for LLM finetuning using multiple languages for generation.

Overall, I like the idea, but I have some comments provided below.

**Strengths:**

The novel data augmentation method.

**Weaknesses:**

The authors compare their method to some unnamed but presumably simplistic technique for data augmentation, I think that it is better to compare to several existing techniques, e.g. starting with classic backtranslation. I suppose that the results could be explained by data augmentation itself, not the language diversity.

**Questions:**

Please give more details with examples on your baseline data augmentation.

---

> ### Author Response · Authors · 2025-11-27
>
> W1: Simplistic data augmentation methods, including classical back-translation or merely translating into multiple languages at the surface level, cannot replicate the gains achieved by our approach. Our empirical results strongly indicate that performance improvements stem primarily from explicitly introducing multilingual diversity into the reasoning process, rather than merely increasing the volume of training examples. Furthermore, our method significantly enhances token efficiency. The results are summarized below. For additional details, please refer to Appendix B.5.
>
> | Method                      | AIME24 (Acc / Tokens)    | GPQA (Acc / Tokens)     | MATH500 (Acc / Tokens)    | Avg (Acc / Tokens)    |
> |----------------------------|--------------------------|--------------------------|---------------------------|-----------------------|
> | multilingual-training (ours) | **0.4333 / 12998.355**   | **0.5303 / 8622.477**    | **0.8960 / 4221.365**     | **0.6198 / 8614.06**  |
> | thinking-en-backtrans      | 0.3667 / 14055.419       | 0.5404 / 11531.000       | 0.8660 / 5131.263         | 0.5910 / 10239.22     |
>
> Q1:Thank you for your suggestion; we will include detailed case examples in the appendix B.4.
>
> In the 8×8 grid path-counting problem AIME24-I-6, the bilingual reasoning version initially proceeded entirely in English and incorrectly assumed the number of direction changes equals the number of switches minus one. At a key moment, it abruptly switched to Chinese, explicitly stating "等等，不对，让我再想一想" ("Wait, that's incorrect, let me rethink it"), and carefully reconsidered the definition with concrete examples like R U R U. This allowed the model to reconstruct the correct logic of dividing the path into five alternating segments RURUR or URURU and applying the combinatorial "stars and bars" method to correctly reach the answer of 294. In contrast, the purely English version repeatedly used "Wait..." but remained stuck in its incorrect logic, counting only half the cases starting with R, totaling 147 paths, and overlooking the symmetric cases. This clearly demonstrates that the Chinese interjection "等等" effectively triggered structural reconsideration and correction, whereas the English "Wait" did not.  For additional details, please refer to Appendix B.4.

---

### Official Review · Reviewer_Gegq · 2025-11-04

**Soundness:** 3
**Presentation:** 2
**Contribution:** 3
**Rating:** 6
**Confidence:** 3

**Summary:**

This paper describes a new method for improving the reasoning efficiency of LLMs at test time via multilingual data. Unlike prior work which focused on single-language CoT annotations and large amounts of fine-tuning data, the authors propose L_2 a multilingual unification learning approach that augments a small number of high-quality CoT examples into multiple languages and mixes language-step reasoning to exploit diverse reasoning patterns across languages. They design experiments by fine-tuning a base model (e.g. Qwen2.5-32B) on only a handful 6 up to ~1000 multilingual annotated samples, evaluating on datasets like AIME24, GPQA-Diamond and MATH500. Experiments show that multilingual augmentation significantly improves reasoning accuracy and also reduces inference token usage compared to monolingual baselines.

**Strengths:**

1. The idea of leveraging multilingual reasoning diversity (rather than simply more data) to increase reasoning efficiency is novel and sound.
2. The authors showed strong empirical gains even with extremely small annotated sample sizes when augmented via multilingual CoT.
3. Addresses both data efficiency and inference efficiency which is increasingly important in practice for LLM deployment.

**Weaknesses:**

1. The experiments use relatively small and controlled benchmark sizes e.g., AIME24 with only 30 problems. It's not very clear how much it would scale to broader diverse reasoning tasks.
2. While the reduction in inference tokens is claimed, detailed breakdowns of token savings vs accuracy trade-offs (e.g. across languages and varying lengths) are not sufficiently discussed.

**Questions:**

1. How sensitive is the method to the quality of the multilingual translations or CoT annotations in non-audited languages?
2. Does the multilingual mixing approach generalize to non-math reasoning tasks?

---

> ### Author Response · Authors · 2025-11-27
>
> W1: For fair comparison, we largely follow the settings in the polular baseline works [1-3]. Although AIME24 is ralatively small, they are very challenging for the advanced large reasoning models. It is non-trivial to curate difficult reasoning problems, so AIME24 (30), AIME25(30), GPQA-D (198) and MATH500 (500) are widely adopted together to balance the difficulty and size.
>
> W2:  We conducted extensive experiments to analyze the trade-offs between token efficiency and accuracy across various multilingual settings (see Section 4.5.2 and Figure 5). Specifically, using the dataset that spans nine languages grouped into three language families—East-Asian (zh/ja/ko), Indo-European (en/fr/de/ru), and Afro-Asiatic (ar/he)—we systematically constructed different multilingual combinations, including single-family sets (e.g., zh/ja/ko), two-family mixtures (e.g., zh/en), and three-family mixtures (e.g., zh/en/he), each used to train separate models. Our experimental results consistently demonstrate that as the diversity of language families in training increases, models simultaneously achieve higher accuracy and lower inference token counts. This trend is visually illustrated in Figure 5, where points progressively cluster toward the upper-left region (indicating higher accuracy with fewer tokens)
>
> Q1: Our method demonstrates robustness against varying quality of multilingual translations and CoT annotations, it does not strictly require annotations to be entirely accurate. Even when annotation accuracy dropped to 30%, performance remained strong. Specifically, we used  LLM as judge approach, validating translations and reasoning steps if they matched the original CoT from S1. We designed three datasets (LabelAcc-30%, 60%, 100%), each with identical sizes, to empirically verify sensitivity. For additional details, please refer to Appendix B.6.
> | Dataset       | AIME24 | GPQA   | MATH500 | Average |
> |--------------|:------:|:------:|:-------:|:-------:|
> | LabelAcc-30%  | 0.4333 | **0.5909** | **0.908** | 0.644 |
> | LabelAcc-60%  | **0.5** | 0.5546 | 0.902 | **0.652** |
> | LabelAcc-100% | 0.4667 | 0.5404 | 0.904 | 0.637 |
> These results confirm our approach's insensitivity to annotation accuracy variations.
>
> Q2: Yes, our approach can generalize to broader reasoning tasks. However, our method assumes different languages might inherently prompt diverse reasoning perspectives, enhancing data diversity. If multilingual reasoning diversity is limited for certain tasks, we acknowledge the potential reduction in benefits from our multilingual augmentation. Nonetheless, our method does not impose constraints regarding whether the reasoning tasks have standard answers (like mathematics) or are open-ended.
> References:
>
> [1] S1: Simple test-time scaling.
>
> [2] Bespoke-Stratos: The unreasonable effectiveness of reasoning distillation.
>
> [3] LIMO: Less is More for Reasoning.

---

### Author Response · Authors · 2025-12-04
**Global Revision Summary**

We sincerely appreciate the reviewers’ detailed and constructive feedback. Below, we summarize the major changes and improvements made in the revised manuscript:

1. **Enhanced Analysis on Efficiency and Multilingual Performance (R-Gegq, R‑2NfV)**:

   * We expanded the empirical analysis of token efficiency vs. accuracy across diverse multilingual scenarios (Section 4.5.2, Figure 5). This clearly demonstrates improved efficiency with increasing language diversity.
   * Clarified the relationship between token usage and other efficiency metrics (e.g., inference time, FLOPs, KV-cache usage), emphasizing the robustness of our token-based metric (Section 4.7).

2. **Additional Data Augmentation Baselines and Ablation Studies (R‑AVEt)**:

   * Added comparisons to a standard back-translation baseline, showing our multilingual training method achieves superior performance with fewer tokens (Appendix B.5, Table 5).
   * Included robustness analysis against annotation noise, illustrating strong performance even when multilingual annotations have lower accuracy (Appendix B.6, Table 6).

3. **Broader Multilingual Evaluations (R‑TAHy, R‑2NfV)**:

   * Provided extensive results on the PolyMath benchmark (18 languages across four difficulty levels), demonstrating substantial gains over baseline models, thus confirming our method’s general effectiveness beyond original benchmarks (Appendix B.7, Tables 7–8, Figure 12).
   * Clarified distinctions between our multilingual unification learning approach and related work, highlighting the unique training-phase integration of multilingual reasoning.

4. **Detailed Investigations on Data Scale and Language Mixing Effects (R‑Gegq, R‑2NfV, R‑TAHy)**:

   * Conducted experiments varying dataset sizes, identifying a clear threshold (~30 samples) where multilingual data significantly boosts model performance and reasoning depth (Section 4.5.1, Figure 4).
   * Demonstrated orthogonality with existing data curation methods, confirming consistent improvements in reasoning accuracy and token efficiency regardless of initial data selection strategy (Section 4.6, Table 3).

5. **Clarified Decoding Intervention and Qualitative Analyses (R‑Gegq, R‑TAHy)**:

   * Clearly defined and systematically tested our decoding-time language intervention approach, showing we can effectively control reflective reasoning language without significantly affecting token usage (Section 3.4, Appendix B.3, Table 4).
   * Added concrete qualitative examples, including detailed multilingual vs. monolingual reasoning cases, clearly illustrating how multilingual reasoning triggers improved reflection and error correction (Section 5, Appendix B.4).

6. **Improved Clarity and Organization (All Reviewers)**:

   * Introduced clearer dataset naming conventions and restructured key sections (especially Section 3.1) to enhance readability and reduce confusion.
   * Revised the manuscript throughout to address presentation concerns, aiming for greater clarity and coherence.

---

### Meta-Review · Area_Chair_omVH · 2026-01-09

**Summary:**

This paper received four mixed reviews, 2 positive recommendations and 2 negative recommendations. All reviewers agreed that the idea is compelling and the performance improvement seems promising.

The major concerns that affect the recommendation include:
- the scale of benchmarks
- missing discussion and comparison with the relevant baseline (Although the authors have claimed that the motivations are different, it would be better to validate that the existing methods fail to work in your scene)
- unconvincing practicability

Overall, I'm inclined to recommend it for a (borderline) rejection.

**Reviewer Concerns:**

The scale of benchmarks.

**Reviewer Scores:**

Most reviewers may prefer to maintain their original score, especially two reviewers with negative ratings. The rebuttal is not strong and not persuasive.

---

### Decision · Program_Chairs · 2026-01-26

Reject